# Serotonergic modulation of visual neurons in *Drosophila melanogaster*

**Maureen M. Sampson**[1,2☯], **Katherine M. Myers Gschweng**[1,3☯], **Ben J. Hardcastle**[4], **Shivan L. Bonanno**[1,3], **Tyler R. Sizemore**[5], **Rebecca C. Arnold**[1,3], **Fuying Gao**[3], **Andrew M. Dacks**[5], **Mark A. Frye**[4], **David E. Krantz**[1,3]*

**1** UCLA, Hatos Center for Neuropharmacology, Los Angeles, CA, United States of America, **2** UCLA, Molecular Toxicology Interdepartmental Program, Los Angeles, CA, United States of America, **3** UCLA, Department of Psychiatry and Biobehavioral Sciences, Semel Institute for Neuroscience and Human Behavior, David Geffen School of Medicine, Los Angeles, CA, United States of America, **4** UCLA, Department of Integrative Biology and Physiology, Los Angeles, CA, United States of America, **5** Department of Biology, West Virginia University, Morgantown, WV, United States of America

☯ These authors contributed equally to this work.
* dkrantz@ucla.edu

**Data Availability Statement:** In addition to the data included in the manuscript, figures, tables and supporting information, the analysis code and the live-imaging data has been deposited at https://osf.

## Abstract

Sensory systems rely on neuromodulators, such as serotonin, to provide flexibility for information processing as stimuli vary, such as light intensity throughout the day. Serotonergic neurons broadly innervate the optic ganglia of *Drosophila melanogaster*, a widely used model for studying vision. It remains unclear whether serotonin modulates the physiology of interneurons in the optic ganglia. To address this question, we first mapped the expression patterns of serotonin receptors in the visual system, focusing on a subset of cells with processes in the first optic ganglion, the lamina. Serotonin receptor expression was found in several types of columnar cells in the lamina including 5-HT2B in lamina monopolar cell L2, required for spatiotemporal luminance contrast, and both 5-HT1A and 5-HT1B in T1 cells, whose function is unknown. Subcellular mapping with GFP-tagged 5-HT2B and 5-HT1A constructs indicated that these receptors localize to layer M2 of the medulla, proximal to serotonergic boutons, suggesting that the medulla neuropil is the primary site of serotonergic regulation for these neurons. Exogenous serotonin increased basal intracellular calcium in L2 terminals in layer M2 and modestly decreased the duration of visually induced calcium transients in L2 neurons following repeated dark flashes, but otherwise did not alter the calcium transients. Flies without functional 5-HT2B failed to show an increase in basal calcium in response to serotonin. 5-HT2B mutants also failed to show a change in amplitude in their response to repeated light flashes but other calcium transient parameters were relatively unaffected. While we did not detect serotonin receptor expression in L1 neurons, they, like L2, underwent serotonin-induced changes in basal calcium, presumably via interactions with other cells. These data demonstrate that serotonin modulates the physiology of interneurons involved in early visual processing in *Drosophila*.

io/39j4m/ (DOI: 10.17605/OSF.IO/39J4M) and the RNA-Seq data (raw and processed files) are available on GEO at https://www.ncbi.nlm.nih.gov/geo/query/acc.cgi?acc=GSE154085.

**Funding:** The work was funded by R01 MH107390 (DEK), R01 MH114017 (DEK) R01 EY026031 (MAF), R03 DC013997 (AMD), R01 DC016293 (AMD) https://www.nih.gov/grants-funding, IOS-1455869 (MAF) https://www.nsfgrfp.org/, and a seed grant from the UCLA Depression Grand Challenge (DEK, MAF) https://grandchallenges.ucla.edu/depression/. MMS was supported by a National Science Foundation GRFP https://www.nsfgrfp.org/, a UCLA Cota-Robles fellowship https://grad.ucla.edu/funding/financial-aid/funding-for-entering-students/eugene-v-cota-robles-fellowship/, and F99 NS113454 https://www.nih.gov/grants-funding. TRS was supported by a Grant-In-Aid of Research (G201410156699888) from Sigma Xi https://www.sigmaxi.org/programs/grants-in-aid The Scientific Research Society. The funders had no role in study design, data collection and analysis, decision to publish, or preparation of the manuscript.

**Competing interests:** The authors have declared that no competing interests exist.

## Author summary

Serotonergic neurons innervate the *Drosophila melanogaster* eye, but it was not known whether serotonin signaling could induce acute physiological responses in visual interneurons. We found serotonin receptors expressed in all neuropils of the optic lobe and identified specific neurons involved in visual information processing that express serotonin receptors. Activation of these receptors increased intracellular calcium in first order interneurons L1 and L2 and may enhance visually induced calcium transients in L2 neurons. These data support a role for the serotonergic neuromodulation of interneurons in the *Drosophila* visual system.

## Introduction

Serotonin acts as a neuromodulator [1–5] in a variety of networks including the sensory systems required for olfaction, hearing, and vision [6–17]. In the mammalian visual cortex, serotonin regulates the balance of excitation and inhibition [6], cellular plasticity [18–21], and response gain [8, 22]. In some cases, the contribution of individual receptor subtypes is known; for example, in the mammalian retina, serotonergic signaling reduces GABAergic amacrine cell input to retinal ganglion cells via 5-HT1A [23]. Additionally, some retinal ganglion cells express 5-HT2C and loss of this receptor enhanced the response to contrast-reversing gratings [24]. However, for most sensory circuits, the manner in which serotonin receptor activation is integrated to regulate their activity and drive adaptive changes remains poorly understood.

The visual system of *Drosophila melanogaster* provides a genetically tractable model to study visual circuit activity and regulation [25,26]. In *Drosophila*, early visual processing occurs in the lamina where intrinsic monopolar neurons receive direct input from photoreceptors [27]. Lamina monopolar cells L1 and L2 are first-order interneurons that feed into pathways discriminating light "ON" (i.e., increase in luminance) and light "OFF" (i.e., decrease in luminance) stimuli respectively [28,29]. L1 and L2 neurons respond to changes in luminance in a physiologically indistinguishable manner [30–32], while downstream neurons in the medulla transform this information to discriminate ON versus OFF stimuli [29]. Further processing occurs in the lobula and lobula plate to mediate higher-order computations for both motion and contrast detection [29,33,34]. Significant progress has been made in mapping the synaptic connectivity and function of visual processing neurons, including those required for motion detection [27,35–40].

Additional studies have shown that aminergic neurotransmitters can modulate visual information processing in flies and other insects [26,41–46]. Octopamine, the invertebrate equivalent of noradrenaline, is present in processes innervating the medulla, lobula and lobula plate in *Drosophila*, where it regulates state-dependent modulation of visual interneurons [42,46] including the saliency of objects during flight [43]. Serotonergic neurons also innervate the optic ganglia [47–52] and previous studies indicate that serotonin impacts cellular activity and visual behaviors in insects [44,45,53–56]. In *Drosophila*, serotonin was shown to modulate the voltage dependence of potassium channels in photoreceptors [44]. In the blowfly, serotonin alters electrophysiological field recordings representing the combined output of lamina neurons [53]. In the honeybee, single cell recordings in motion-sensitive lobula neurons showed that serotonergic signaling reduces background activity, directional selectivity, and the amplitude of field potentials evoked by moving stripes [45].

The visual system is also essential for setting and maintaining circadian rhythms [57,58]. Physiological changes in photoreceptors and downstream interneurons may facilitate adaptation to circadian changes in light intensity or other stimuli. In houseflies, serotonin and other neurotransmitters regulate daily rhythmic swelling of monopolar cell L2 terminals in the medulla [59], a process that was proposed to decrease monopolar signal in the daytime when light levels increase. In *Drosophila*, serotonin levels decrease in constant darkness [56], possibly correlating with changes in photosensory input and contributing to adaptation to varying light levels throughout the day. In contrast, serotonin levels increase during the dark phase in the sphinx moth [9] and cricket [60]. Serotonin also regulates circadian behaviors in *Drosophila* [55,56], but the potential contributions of specific subtypes of serotonin receptors to circadian processes or more acute changes in the visual environment remains unclear.

Serotonin receptor signaling occurs via diverse secondary messenger cascades [61,62] and receptors may act individually or in combination within a single cell [63,64] or circuit [65]. Co-activation of both 5-HT1A and 5-HT2A in cortical pyramidal neurons leads to complex physiologic responses [65,66] that are further regulated by 5HT1A receptors on local inhibitory neurons [67]. The effects of receptor activation can also vary depending on inputs from other neuromodulators [68] and the differential activation of downstream effectors [65,69]. These studies highlight the difficulties of predicting the effects of serotonin receptor activation based solely on expression data, and the need for physiological assays to assess their role in specific circuits. Although two recent studies have reported serotonin receptor expression in visual neurons [70,71], to our knowledge there is no information on the physiological effect(s) of any serotonin receptor in any of the interneurons within the insect visual system. In this work, we show how a specific serotonin receptor, 5-HT2B, affects intracellular calcium and visual responses in L2 lamina monopolar cells, which are critical for early visual information processing.

## Results

### Distinct lamina neurons express different serotonin receptors

Five genes encoding serotonin receptors have been identified in the *Drosophila* genome: 5-HT1A, 5-HT1B, 5-HT2A, 5-HT2B and 5-HT7 [72–76]. To identify specific optic lobe neurons expressing each receptor, we expressed the marker mCD8::RFP (or GFP) under the control of a recently characterized panel of T2A-GAL4 insertions in Minos-Mediated Integration Cassettes (MiMICs) located in serotonin receptor gene introns [77]. The GAL4 sequence is inserted into receptor-encoding genes where it acts as an artificial exon and is expected to "mimic" the endogenous gene expression patterns [78]. Ribosome skipping via T2A allows GAL4 to be expressed as a separate protein, rather than a fusion protein with the serotonin receptor [77,79].

We observed distinct expression patterns for each receptor including projections into the optic lobe neuropils: the lamina (la), medulla (me), lobula (lo) and lobula plate (lp) (S1 Fig). We focus here on receptor subtypes showing expression in the lamina because of the ease of identifying cells based on their morphology [80], the established role of some lamina neurons in the response to experimentally tractable visual stimuli [28,29,81], and the proposed relationship between serotonin and circadian changes in a subset of lamina neurons in larger flies [59].

To identify the specific cell types that express each serotonin receptor in the lamina, we used the receptor MiMIC-T2A-GAL4 lines described above in combination with the sparse labeling technique MultiColor FlpOut 1 (MCFO) [82]. Using 5-HT1A and 5-HT1B MiMIC-T2A-GAL4 lines with MCFO we frequently observed a cell with a soma in the medulla cortex,

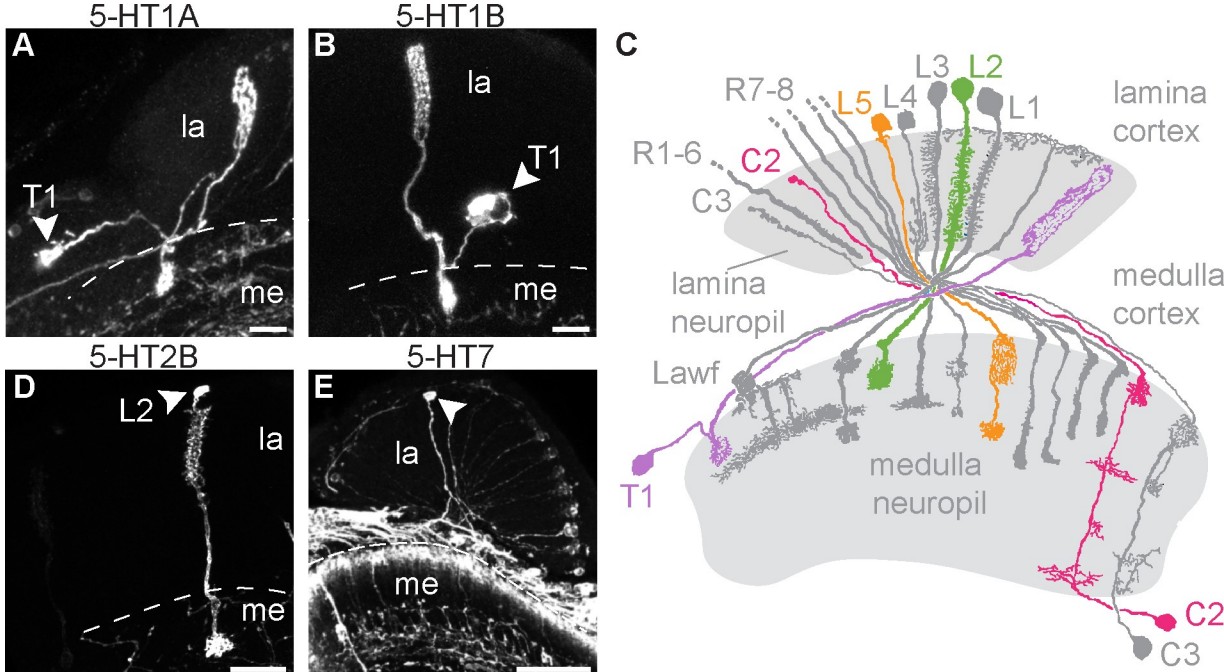

**Fig 1. Lamina neurons including T1 and L2 express serotonin receptors.** (A-E) Serotonin receptor MiMIC-T2A-GAL4 lines were crossed to UAS-MCFO-1 to sparsely label individual cells in the lamina. Cell bodies are indicated by an arrowhead. 5-HT1A (**A**) and 5-HT1B (**B**) MCFO crosses revealed cells with morphologies identical to T1 neurons. (**C**) A diagram showing lamina neurons adapted from [80] highlights L2 (green), L5 (orange), T1 (purple), and C2 (pink). (**D**) 5-HT2B>MCFO labeled cells were morphologically identical to L2 neurons. (**E**) 5-HT7>MCFO-1 labeled neurons possibly representing L5 lamina monopolar cells. Scale bars are 20 μm and N = 9–31 brains imaged per receptor subtype. Due to the nature of stochastic labeling, some cell types were observed in only a subset of brains: 22/31 (**A**), 10/11 (**B**), 9/9 (**D**), and 7/13 (**E**).

a long basket-like projection in the lamina, and a smaller projection in the medulla (Fig 1A and 1B). This morphology is identical to that of T1 cells and distinct from other cell types in the lamina (Fig 1C) [80]. T1 cells were labeled in 23 of 31 brains (71%) for 5-HT1A and in 10 of 11 brains (91%) for 5-HT1B. On average, we observed thirteen MCFO-labeled T1 cells per individual optic lobe for 5-HT1A and nine T1 cells per optic lobe for 5-HT1B. These data are consistent with the results of recently published studies that used TAPIN-Seq or FACS-S-MART-Seq to analyze expression in T1 as well as other cells in the visual system [70,71] (see S2 Fig for a comparison of these findings).

Using the 5-HT2B MiMIC-T2A-GAL4 driver with MCFO, we observed cells with a soma in the lamina cortex, dense projections extending into the lamina neuropil, and a single bushy terminal in the medulla (Fig 1D), together representing a morphology identical to lamina monopolar neuron L2 and no other lamina cell types (Fig 1C) [80]. We observed L2 cells in 9 of 9 (100%) 5-HT2B>MCFO brains, observing an average of eleven L2 neurons per optic lobe. Additionally, we co-expressed 5-HT2B>RFP with MiMIC-T2A-Lex-ChAT>GFP and found that a subset of the lamina monopolar cells co-labeled with both lines, consistent with the neurochemical identity of L2 cells as cholinergic (S3A Fig, arrowheads).

For 5-HT7>MCFO, we observed lamina monopolar cells in 7 of 13 brains (54%) (Fig 1E), with an average of 20 cells per optic lobe. Over 99% of the lamina monopolar cells labeled with 5-HT7>MCFO lacked the dense processes in the lamina neuropil that are characteristic of L1-L3, and also lacked the vertically oriented collaterals in the inner (proximal) lamina seen in L4 neurons (Fig 1C and 1E). We therefore suggest that 5-HT7 may be expressed in the one

remaining lamina monopolar cell subtype, L5. Additional examples of 5-HT7>MCFO labeled cells possibly representing L5 are included in S3B and S3C Fig.

We detected additional, less frequent events for other columnar cells in the lamina. These include observations of multiple centrifugal cells (C2 or C3) in 3 of 31 5-HT1A>MCFO brains (S3D Fig) consistent with two previous reports of 5-HT1A in C2 [70, 71]. In further support of the expression of 5-HT1A in centrifugal cells, we observed co-expression of 5-HT1A>RFP and MiMIC-LexA GAD1>GFP in thin projections in the lamina and in cell bodies between the proximal medulla cortex and the lobula plate (S2E Fig, arrowheads).

## Medulla neurons, glia and serotonergic neurons express serotonin receptors

The results of other more comprehensive studies of RNA expression in visual system neurons prompted us to look beyond our primary focus of lamina neurons [70,71]. In addition to the columnar neurons that express serotonin receptors and extend projections into the lamina [80], we have tentatively identified cells with projections that are either confined to the medulla, or include both medulla and lobula complex, and may also express serotonin receptors (S3F–S3H Fig).

For 5-HT2A>MCFO, additional pleomorphic labeling was observed in the lamina cortex (S4A Fig) in a pattern that appeared similar to that of optic lobe glia [83]. Anti-repo labeled nuclei showed close proximity to many 5-HT2A labeled cells (S4B Fig), but one-to-one matching was not possible due to irregular cell morphology. A previous microarray study of glia suggested that 5-HT1A and 5-HT7 are enriched in repo-GAL4 specified glia, while 5-HT1B was enriched in surface glia [84]. Another previous study [70] reported expression of 5-HT7 in three types of lamina glia—epithelial glia, proximal satellite glia and marginal glia.

The major ganglia of the visual system do not contain any serotonergic cell bodies; rather projections from neurons in the accessory medulla and central brain innervate the optic lobes [50,85–88]. We observed a cluster of 8–10 cell bodies in the accessory medulla (S5A Fig, S5C Fig) that corresponds to cluster LP2 (or Cb1), which was previously reported to project into the medulla [85]. Additionally, two symmetric, serotonin-immunoreactive neurons designated LBO5HT have cell bodies in the ventral protocerebrum and bilaterally innervate the lamina, medulla, lobula and lobula plate [85]. Immunolabeling for serotonergic boutons can be observed within all optic ganglia neuropil as well as the lamina cortex (S5A Fig) [50,85–88]; e.g. arborizations from 5-HT1B>GFP labeled cells in the outer medulla, which we have identified as T1 projections, were surrounded by a honeycomb pattern of serotonergic immunolabeling (S5C Fig).

Sparse labeling with 5-HT1B>MCFO co-labeled with serotonin-immunolabeled boutons in the inner medulla (iM), medulla layer 4 (M4), and lobula (lo) (S5E Fig), suggesting that 5-HT1B could function as an autoreceptor in these processes. Previous studies have also reported 5-HT1B and 5-HT1A as potential autoreceptors in *Drosophila* [56,89]. Although we did not comprehensively map all putative serotonin autoreceptors in the central brain, we used serotonergic cell maps described in [50,85,87,88] to identify 5-HT1B+ cell clusters as LP2 (Cb1), PLP (LP1), and PMP, (S5 Fig) and 5-HT1A+ serotonergic clusters as PLP (LP1), SEL, AMP and PMP (S6 Fig). The integration of physiological studies in both serotonergic neurons and post-synaptic neurons represents a future goal to assess the interplay between auto- and post-synaptic receptors in visual circuits.

## L2 neurons express 5-HT2B and T1 neurons express 5-HT1A and 5HT1B

We sought to confirm the expression of serotonin receptors independently of both previous studies [70,71] and the data we obtained using MiMIC-T2A-GAL4 lines. To this end, we used

a separate set of split-GAL4 [90] or LexA drivers previously shown to be specific for particular cell types, and focused on a small subset of lamina neurons: T1, L1 and L2. Drivers representing each cell were used to express GFP, and the GFP-labeled cells were isolated via Fluorescence Activated Cell Sorting (FACS). RNA was then extracted from T1, L1 and L2 GFP + FACS isolates as well as the unlabeled cells. To probe for serotonin receptor expression in each cell type, we used both RNA-Seq (Fig 2 and S1 Table) and RT-qPCR (S7 Fig and S2 Table).

For RNA-Seq, we compared the relative abundance for each receptor by calculating Transcripts Per Kilobase Millions (TPMs). We found that 5-HT1A (182±43 TPM±std) and 5-HT1B (278±25 TPM±std) were more abundant than other serotonin receptors (range 0.01 ±0.01 to 2.5±4 TPM±std) in T1 samples (N = 3, Fig 2A and S1 Table). Two previous sequencing studies similarly reported the expression of 5-HT1A and 5-HT1B in T1 [70,71]. L2 isolates showed higher TPMs for 5-HT2B (130±75 TPM±std) compared to other serotonin receptors (range 6±7 to 31±9 TPM±std) (Fig 2B and S1 Table), also consistent with the results of another recent transcriptomic study [70]. For RT-qPCR, we calculated enrichment (i.e., fold change) relative to pooled, unlabeled optic lobe cells using the comparative CT method [91]. Relative to unlabeled optic lobe neurons, L2 samples were enriched for 5-HT2B in 4/5 samples but also showed enrichment for 5-HT7 and 5-HT2A in 3/5 and 1/5 samples respectively (S7 Fig and S2 Table). Although we cannot rule out the presence of 5-HT2A or 5-HT7 in L2 based on the results of RT-qPCR, our data and those of others [70] suggest that 5-HT2B may be the only serotonin receptor abundantly expressed in L2 cells.

We did not observe evidence of any serotonin receptor expression in L1 neurons using the serotonin receptor MiMIC-T2A-GAL4 lines to drive MCFO. In agreement with this observation, RT-qPCR from isolated L1 cells (S7 Fig and S1 Table) showed virtually no receptor expression, apart from one sample weakly enriched for 5-HT1B. Others have also reported a low likelihood of any serotonin receptor expression in L1 neurons [70]. In sum, MCFO sparse labeling in combination with RNA-Seq and RT-qPCR by our group and others [70,71] indicate that T1 neurons express 5-HT1A and 5-HT1B, and L2 neurons express 5-HT2B, whereas L1 neurons may not express any serotonin receptor subtypes.

## 5-HT2B and 5-HT1A receptors localize to the medulla neuropil

Both T1 and L2 neurons have dense projections in the lamina neuropil and arborize in layer 2 (M2) of the medulla neuropil. Serotonergic neurons directly innervate M1 and M2 of the medulla neuropil raising the possibility that serotonergic signaling might occur at this site. If so, we reasoned that the serotonin receptors expressed in L2 and T1 might localize to M1 and/or M2. To test this possibility, we took advantage of a 5-HT1A allele that had been tagged at the C-terminus with GFP [92]. The tag was inserted into the endogenous 5-HT1A gene, such that, similar to MiMIC-T2A-GAL4 lines [77, 78], the receptor::GFP fusion protein product is putatively expressed at the same level and in the same cells as the endogenous protein [92]. In 5-HT1A::GFP flies, we observed enrichment of the tagged protein in layer M2 of the medulla relative to other subcellular sites (Fig 3A–3A"), suggesting that serotonergic signaling to 5-HT1A::GFP expressing neurons occurs in this region. This might include T1 as well as other columnar neurons that extend processes into M2. Regardless of cell type, there is very low 5-HT1A::GFP signal in the lamina neuropil (Fig 3A'), suggesting that serotonergic signaling primarily targets 5-HT1A receptors localized to the medulla neuropil. We also observed anti-serotonin immunoreactive puncta that co-labeled with 5-HT1A::GFP in the medulla suggesting that 5-HT1A could act as an autoreceptor in serotonergic projection neurons innervating the medulla (Fig 3B, arrows, and S6 Fig).

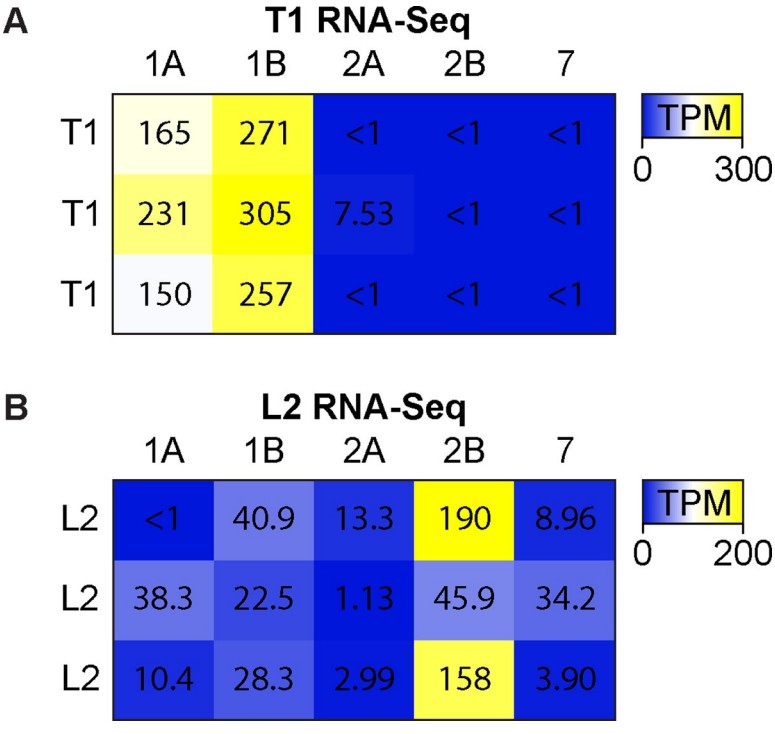

**Fig 2. L2 neurons express 5-HT2B and T1 neurons express both 5-HT1A and 5-HT1B serotonin receptors.** T1 or L2 neurons were isolated by FACS for RNA-Seq. Transcript abundance was calculated as Transcripts Per Kilobase Million (TPMs). (**A**) T1 RNA-Seq TPMs (color low to high, blue to yellow) were higher for 5-HT1A and 5-HT1B compared to other serotonin receptors. (**B**) L2 RNA-Seq showed high abundance 5-HT2B in two of three samples, with 5-HT2B being the most abundant serotonin receptor transcript in all replicates.

Since we were unable to obtain an endogenously tagged allele of 5-HT2B, we relied on expression of the UAS-5-HT2B::sfGFP transgene [93] under the control of L2-split-GAL4 to investigate its subcellular localization (Fig 3C). 5-HT2B::sfGFP was enriched in terminals within M2 of the medulla as compared to the lamina neuropil (Fig 3C–3C") and showed additional punctate labeling in L2 cell bodies (Fig 3C', arrowhead). To control for the possibility that all membrane-bound proteins might appear to be enriched in M2, we expressed the plasma membrane marker UAS-mCD8::GFP using the same L2-split-GAL4 driver. In contrast to 5-HT2B::sfGFP, labeling with mCD8::GFP was most prominent in the cell body and proximal processes with progressively weaker labeling through the lamina and medulla neuropil and no enrichment in layer M2 (Fig 3D"). These data strongly suggest that both 5-HT2B and 5-HT1A preferentially localize to the terminals of L2 and T1 respectively in the medulla layer M2 rather than the lamina neuropil. Serotonergic boutons also localize to several layers within the medulla neuropil but are not found in the lamina neuropil (see S5A Fig). It is therefore more likely that T1 and L2 neurons receive serotonergic signals in the medulla rather than the lamina. This may occur in M2, although we cannot rule out other sites in the medulla where serotonergic boutons are present, but the receptors are less enriched.

To further explore serotonergic signaling to L2 and T1 in the medulla, we used sybGRASP to probe for potential synaptic connections between serotonergic boutons in M2 and the terminals of L2 and T1 [94]. We used the previously established ultrastructural connectivity of L2 onto T1 neurons in the medulla [37,38] as a positive control to validate the use of sybGRASP in detecting interactions within M2, and obtained a robust signal (S8A Fig). By contrast, we did not detect a signal in M2 in sybGRASP experiments in which the serotonergic neurons

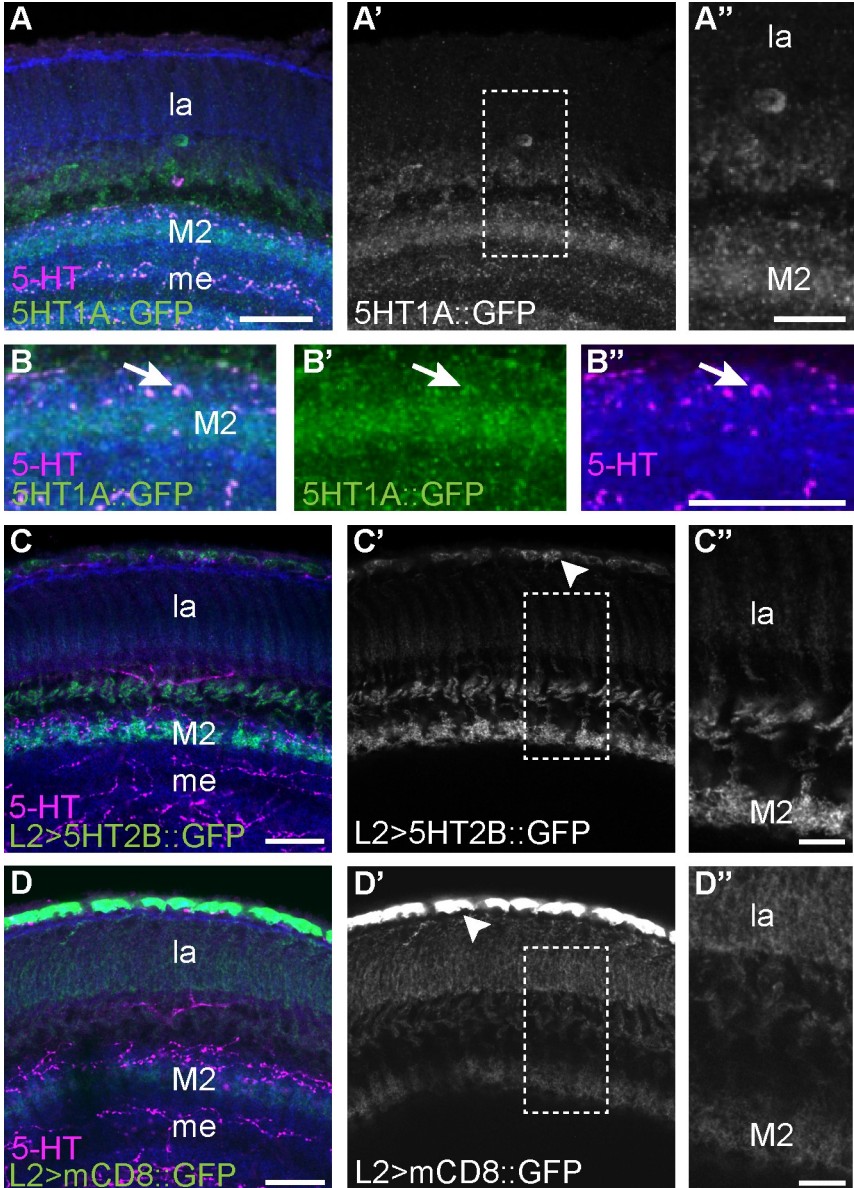

**Fig 3. Serotonin receptors 5-HT1A and 5-HT2B are enriched in layer M2 of the medulla.** (**A-A"**) 5-HT1A::GFP localized to layer M2 of the medulla, adjacent to some of the boutons in the medulla neuropil immunogenic for serotonin. Neuropil (anti-N-Cadherin, blue) and serotonin (magenta) labeling provide anatomical context for the lamina (la) and medulla (me). The region specified by the dotted lines in (**A'**) is enlarged in (**A"**). (**B-B"**) In some cases, serotonin immunoreactive boutons appeared to be co-labeled rather than adjacent to 5-HT1A::GFP (arrow) possibly representing projections from serotonergic neurons in the central brain. (**C-C"**) Subcellular localization of 5-HT2B was visualized by expressing the fusion construct UAS-5-HT2B::sfGFP in L2 neurons specified by L2-split-Gal4. L2>5-HT2B::sfGFP labeling was stronger in the L2 terminals in medulla layer 2 (M2) compared to L2 projections in the lamina (la). Punctate GFP signal was observed in the L2 cell bodies (arrowhead). The region specified by the dotted lines in (**C'**) is enlarged in (**C"**). (**D-D"**) For comparison, membrane-directed UAS-mCD8::GFP was expressed in L2 neurons. L2>mCD8::GFP signal was similar in the medulla and lamina compartments. Strong GFP signal was observed in the cell bodies (arrowhead). The area within the dotted lines in (**D'**) is enlarged in (**D"**). Scale bars are 20 μm in **A-A'**, **B-B"**, **C-C'** and **D-D'**. Scale bars are 10 μm in **A"** and 8 μm in both **C"** and **D"**. Biological replicates are N = 6 for 5-HT1A::GFP (**A-B**), N = 15 for L2>5-HT2B::sfGFP (**C-C"**), and N = 10 for L2>mCD8::GFP (**D-D"**).

were "presynaptic" to L2, T1 or L1 (S8B–S8E Fig). These data suggest that serotonergic signaling to L2, T1 (and perhaps other columnar neurons), is more likely to be mediated by volume transmission rather than true synaptic transmission, consistent with the use of volume transmission by most aminergic synapses in mammalian systems [95–98].

### Serotonin increases calcium levels in L2 and L1 neurons

The data presented here and by others [70,71] strongly suggest that L2 and other cells in the visual system express serotonin receptors but do not address their function. To address the potential effects of serotonin on L2 neurons, we bath applied serotonin to the optic lobe and used live imaging to monitor cellular activity. The concentration of serotonin (100 μM) was within the range used in other *Drosophila* studies [89,99,100]. The data for receptor expression in L2 was strongest for 5-HT2B receptors, which couple with the $G_{q/11}$ protein alpha subunit to increase intracellular calcium *in vitro* [76,101]. We therefore used the genetically encoded calcium indicator GCaMP6f [102] to follow changes in L2 activity that we hypothesized could be induced by serotonin. We again employed the L2-split-GAL4 driver used for transcriptional analysis (Fig 2) to specifically express GCaMP6f in L2 neurons (Fig 4A). Since we observed enrichment of 5-HT2B::sfGFP in L2 terminals in M2, we focused our recordings of calcium signaling on these sites. For each experiment, we first recorded a baseline while perfusing the tissue with saline; the perfusion solution was then switched to either saline containing 100 μM serotonin or saline alone. Throughout the experiment, flies were exposed to a constant low-level luminance to control for visual input and tetrodotoxin (TTX) was included in the perfusion solution to reduce inputs to L2 neurons. In *Drosophila*, TTX inhibits the only known sodium channel $DmNa_V$ (formerly *para*) and represents a standard method to reduce neuronal inputs [99,103,104].

We consistently observed a large increase in GCaMP6f fluorescence in L2 terminals following serotonin application (Fig 4B and S9A Fig). This increase continued throughout the time course of the recordings, peaking at 1.73 ΔF/F ± 0.77 SEM (compared to saline control -0.03 ΔF/F ± 0.05 SEM at the same timepoint; p = 0.0095 by two-tailed Wilcoxon rank sum test). Thus, serotonin leads to an accumulation of cytosolic calcium in L2 cells, consistent with the predicted outcome of activating $G_{q/11}$-coupled 5-HT2B receptors [76,101].

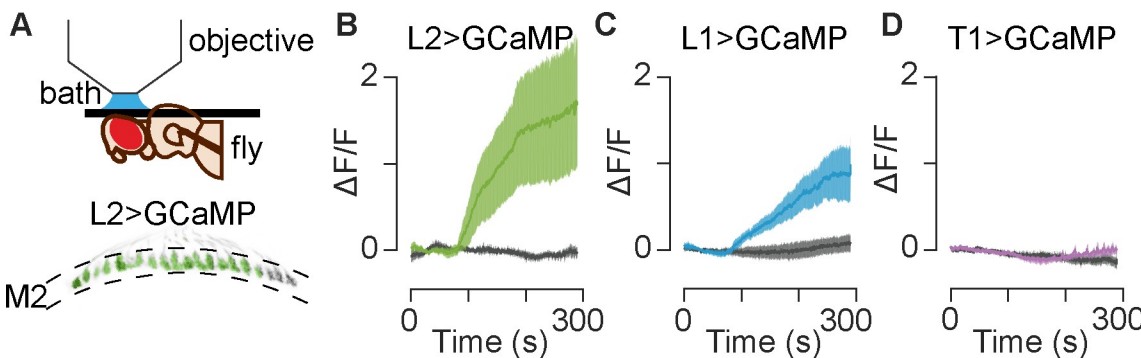

**Fig 4. Bath application of serotonin leads to increased calcium in L2 and L1 neurons, but not T1 neurons.** (**A**) The experimental setup is shown in the top of panel A, along with a sample image of L2 terminals (bottom of panel A, gray) as imaged in the medulla. The ROIs are overlaid in green. (**B-D**) GCaMP6f was paired with L2, L1 and T1 split-GAL4 drivers to monitor responses to 100 μM serotonin (colored traces) or saline controls (gray traces). The perfusion change occurred approximately 105 s into the recording, which corresponds to 45 s in (B-D) because the first 60 s are not shown. (**B**) In L2 terminals, serotonin application led to a significant increase in GCaMP6f signal indicating increased calcium levels as compared to saline controls (p = 00095). (**C**) L1 terminals showed a similar increase in calcium following a switch to serotonin (p = 0.02). (**D**) T1 cells expressing GCaMP6f showed no significant change in calcium following serotonin application (p>0.05). For (B-D) N = 4–8 individual flies; the dark trace is an average of all traces and the shaded region is 1 SEM; saline vs. serotonin comparisons are two-tailed Wilcoxon rank sum tests.

Since we did not detect serotonin receptors in L1 neurons, we did not expect serotonin to measurably change intracellular calcium levels in these cells. However, with GCaMP6f expressed in L1 cells using a cell-specific driver, we regularly observed a robust increase in fluorescence following serotonin exposure (Fig 4C and S9B Fig). Although the increase in GCaMP6f signal did not reach the same response amplitude as observed in L2 neurons, the time course was similar: the signal persisted throughout the recording and peaked at 0.98 ΔF/F ± 0.34 SEM (compared to saline control at 0.07 ΔF/F ± 0.09 SEM; p = 0.02 by two-tailed Wilcoxon rank sum test). A direct action of serotonin on L1 is unlikely since neither we (S7C Fig) nor others [70] detect endogenous serotonin receptors in L1 neurons. Possible mechanisms include inputs from either columnar or non-columnar neurons intrinsic to the visual system, more distal projections from the central brain, glial interactions or perhaps electrical coupling between L1 and L2 [28]. However, TTX was included in the perfusion solution in these experiments to reduce neuronal inputs to L1 including those mediated by action potentials and any graded potentials dependent on the *Drosophila* sodium channel $DmNa_V$ (formerly *para*) [103,104].

We next examined whether serotonin could affect the activity of T1 cells. Both 5-HT1A and 5-HT1B receptors, expressed in T1 neurons, are expected to couple with $G_i$ proteins and negatively regulate adenylyl cyclase [74,101]. Due to the generally inhibitory function of these receptors, we hypothesized that serotonin would dampen activity in T1 neurons, possibly manifested as a decrease in cytosolic calcium or membrane potential [105]. Using the T1 split-GAL4 driver [90] to express either GCaMP6f or the voltage sensor Arclight [106], we did not observe a significant change in fluorescence during perfusion with serotonin (Fig 4D, S9C and S9D Fig) p>0.1). Thus, further experiments will be needed to determine the effects of serotonin on T1 neurons. These negative data are nonetheless important for the current study, since the absence of a GCaMP6f response in T1 neurons indicates that the responses observed in L1 and L2 are not artifacts or a generalized phenomenon common to all cells in the lamina.

## Serotonin in visual processing

To explore the possibility that serotonergic neuromodulation plays a role in visual processing, we tested whether exogenous serotonin would alter visually induced calcium transients in L2 neurons. We used GCaMP6f to record and compare calcium transients in flies receiving saline or serotonin perfusion. Previous studies found that L2 neurons depolarize in response to dark flashes and hyperpolarize in response to light flashes [31,107]. Similarly, calcium-indicator recordings showed that intracellular calcium increased in the dark and decreased in the light [30,108]. Brief light or dark flashes induce bi-phasic calcium transients [31] that enable analysis of calcium kinetics. For this reason, we used brief dark or light flashes to test whether serotonin might alter the magnitude or kinetics of visually induced calcium transients in L2 terminals.

Flies were suspended over an LED arena (see Fig 4A) and either a light or dark flash of the entire LED screen (100 ms) was presented at 5-s intervals. Between each flash, the screen showed an intermediate brightness level, indicated as grey in Fig 5A. One-minute "epochs" consisting of 12 flashes of randomly shuffled polarity were presented six times for each trial (Fig 5A). The first 60-s epoch was recorded in saline alone, followed by a switch to either saline with 100 μM serotonin or saline alone during epoch 2 (Fig 5A). Unlike the experiments shown in Fig 4, we did not include TTX in these experiments so that we could measure L2 responses without dampening neuronal activity in response to visual stimuli.

Dark flashes induced a large increase in calcium that returned to basal levels within ~1 second as previously described [30,31,108] (Fig 5B). The amplitude of the dark flash-induced

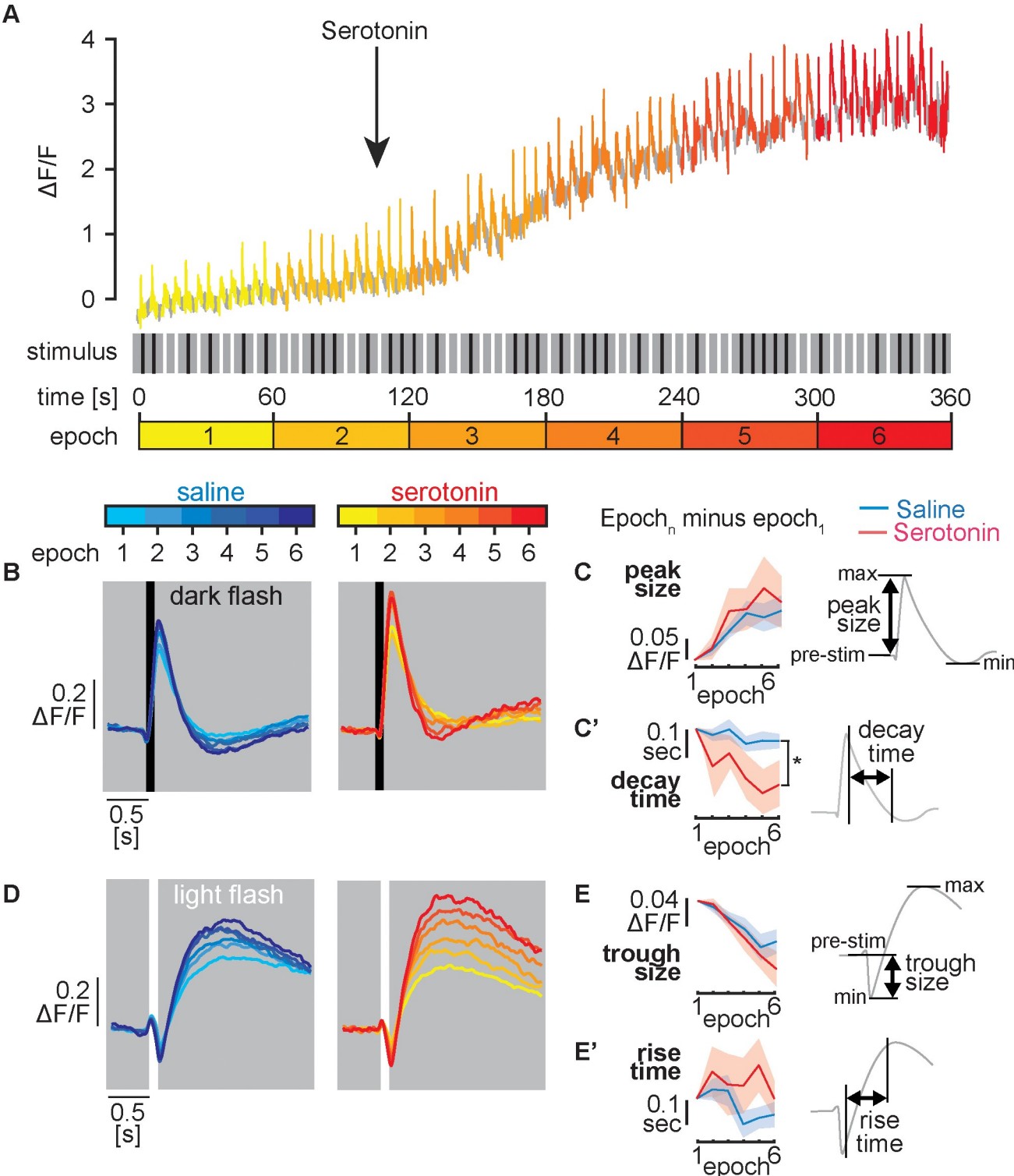

**Fig 5. Serotonin modulates L2 neuron visually induced calcium transient kinetics.** Visually induced calcium transients were recorded in flies expressing L2-split-GAL4>GCaMP6f. For all experiments, a baseline recording in saline (~105 s) was followed by perfusion with either serotonin or saline. (**A**) A sample recording is shown in the upper panel. Light or dark stimuli (A, middle panel, white and black vertical bars respectively) were flashed at random every 5 s for 6 min. To visualize changes in the response to light and dark flashes in L2, data from each 60 s epoch was binned (see epochs 1–6 in A, lower panel). (**B**) Response to dark flashes. Color coded traces representing each 60 s epoch are shown for flies receiving saline (left panel, cyan to blue) or serotonin perfusion (right panel, yellow to red). In both groups, L2 terminals responded to a dark flash with a strong increase in GCaMP6f

fluorescence. (**C-C'**) Analysis of dark flash response from panel B. For plotting each variable shown in C, the average value for the epoch 1 baseline was subtracted (Epoch$_n$-epoch$_1$). Epoch 1 is therefore always set to 0 in panel C-C". (**C**) The change in the calcium transient peak size (the difference between pre-stimulus $\Delta F/F$ and maximum $\Delta F/F$) relative to epoch 1. (**C'**) Change in time (s) from 90% to 10% of the peak-to-peak difference in $\Delta F/F$ (post stimulus max to min), relative to epoch 1. (**D**) Response to light flashes. When a light flash was presented in control experiments, L2 cells responded with a decrease in GCaMP signal followed by a large sustained increase in the GCaMP6 signal. Color coded traces representing each 60 s epoch for saline (left panel, cyan to blue) and serotonin (right panel, yellow to red) are shown. (**E-E'**) Analysis of light flash response from panel D. The average value for the epoch 1 baseline was subtracted as in panel C-C'. (**E**) Change in the calcium trough size (the difference between pre-stimulus $\Delta F/F$ and the subsequent minimum $\Delta F/F$) relative to epoch 1. (**E'**) Change in the time (s) from 10% to 90% of the peak-to-peak difference relative to epoch 1. Recordings in B-E represent, N = 14 and N = 20 individual flies perfused with serotonin or saline respectively. Shaded areas show mean +/- SEM. Comparisons are two-way repeated measures ANOVA with the bracket in 5C' indicating a significant effect of genotype (p≤0.05*).

calcium transients increased over the time course of the experiment for animals receiving either serotonin or saline (Fig 5B). To compare differences between the serotonin and saline control groups and the potential effects of serotonin over the time course of the experiment, we quantified calcium transient peak size (Fig 5C), the time required for decay from 90% to 10% of the peak-to-peak difference (post stimulus maximum minus post stimulus minimum) in $\Delta F/F$ (Fig 5C'), the exponential time constant of the decay (S10B Fig) and the size of the secondary response (S10A Fig). To facilitate the direct comparison of results obtained during perfusion with saline alone versus saline followed by serotonin, we first calculated the average for each variable prior to serotonin exposure measured during epoch 1. We then subtracted the epoch 1 baseline from each subsequent epoch and set the value of epoch 1 in each plot to 0 (indicated as Epoch$_n$-epoch$_1$ in Fig 5C and 5C').

We did not detect a difference between the serotonin and saline groups for changes in peak size (Fig 5C), the exponential time constant of the decay (S10B Fig) or the size of the secondary calcium response (S10A Fig). By contrast, we detected a modest, but statistically significant interaction between time and genotype (p = 0.04 by repeated measures two-way ANOVA) for the decrease in the decay time (Fig 5C').

Light flashes induced a transient decrease in GCaMP6f fluorescence, followed by a secondary calcium increase (Fig 5D) as previously reported [31]. To analyze calcium transients induced by light flashes, we quantified the trough size, defined as the magnitude of the initial calcium decrease relative to the pre-stimulus baseline (Fig 5E), the rise time, defined as the time from 10% to 90% of the peak-to-peak difference in $\Delta F/F$ (Fig 5E'), the exponential time constant of the rise time (S10D Fig) and the magnitude of the secondary response (S10C Fig). Epoch 1 baseline values obtained for the response to light flashes were again subtracted from each epoch to set the initial value of each variable to 0 for epoch 1 in Fig 5E and 5E'. We did not observe any significant differences between saline and serotonin groups when calculating the trough size (Fig 5E), rise time (Fig 5E'), the exponential time constant (S10D Fig) or the magnitude of the secondary response (S10C Fig).

In sum, serotonin drove a robust increase in intracellular calcium levels of L2 neurons in wildtype flies, regardless of the presence (Fig 4) or absence (Fig 5) of TTX. We also detected a modest, but statistically significant effect of serotonin on the decay time of the initial response to a dark flash, but not other metrics, and did not detect any effect of serotonin on the response of L2 to a light flash.

## 5-HT2B mediates the effects of serotonin on L2

Since 5-HT2B is expressed in L2 neurons and is predicted to use calcium as a second messenger, we hypothesized that it would mediate the GCaMP6 response of L2 neurons to serotonin. In flies expressing wild type 5-HT2B we observed a robust serotonin-mediated increase in basal intracellular calcium (Fig 4B and Fig 6A). We next examined whether loss of 5-HT2B would reduce the gradual increase in basal calcium we observed in flies exposed to serotonin

(Fig 6B). As a negative control for experiments using the 5-HT2B homozygous mutants (-/-) we used heterozygous siblings (+/-) in which one wild type allele of 5-HT2B was present (Fig 6B). Similar to 5-HT2B +/+ flies (Fig 6A), 5-HT2B heterozygous controls perfused with serotonin showed a gradual increase in basal calcium levels over the six-minute time course of the experiment (Fig 6B). In contrast, in the 5-HT2B -/- mutant, the basal calcium signal was nearly flat over the time course of the experiment (p = 0.0024 by two-tailed Wilcoxon rank sum test). Although we cannot rule out indirect effects from 5-HT2B expression in other cell types, the simplest explanation for the observed results is that activation of 5-HT2B in L2 neurons generates a gradual increase in cytosolic calcium.

We next examined whether loss of 5-HT2B would alter the calcium transients in L2 neuron terminals following light or dark flashes. As a negative control, we again used heterozygous siblings (5-HT2B +/-). During epoch 1, the period before serotonin perfusion, the magnitude of both the dark-induced calcium increase and light-induced decrease was larger in 5-HT2B +/- compared to 5-HT2B -/- flies (compare Fig 6C and Fig 6E). Further experiments will be needed to determine the origin of this effect and whether it could distort the cells' response to serotonin.

To compare how the loss of 5-HT2B might affect the response to serotonin perfusion, we graphed averages for each epoch after subtraction of the average for epoch-1—the same approach we used in Fig 5C and 5E. As we observed in 5-HT2B +/+ flies (see Fig 5C'), perfusion of both 5-HT2B +/- heterozygous controls and 5-HT2B -/- flies with serotonin led to a gradual reduction in the time required for L2 terminals to reach a minimum after responding to a dark flash (Fig 6D'). The effects on peak size (Fig 6D) were not statistically significant (p = 0.08) between the 5-HT2B -/- mutant and heterozygous controls despite the apparent divergence of the curves. The decay time (Fig 6D and 6D'), size of the secondary response (S11A Fig) and the exponential decay constants (S11B Fig) were also similar, although the final time point for the exponential decay showed a modest difference between the 5-HT2B -/- mutant and heterozygous controls (S11B Fig).

In response to light flashes, the 5-HT2B +/- heterozygote preparations showed a progressive increase in the magnitude of the trough size (Fig 6F); this progression was absent in 5-HT2B -/- mutants. The interaction between the time course and genotype was highly significant (p≤0.0001), the effect of genotype alone was also significant (p = 0.0026) (Fig 6F), and in post hoc tests multiple time points differed between heterozygotes versus homozygotes. Conversely, the rise time (Fig 6F'), the size of the secondary response (S11C Fig) and the exponential time constant (S11D Fig) did not significantly differ between the mutant versus control (Fig 6F'). In sum, the differences between 5HT2B +/- and 5-HT2B -/- in at least one measure of their response to a light flash suggest that 5-HT2B may regulate the effects of serotonin on the response of L2 neurons to some visual stimuli.

## Discussion

To develop the fly visual system as a molecular-genetic model to study serotonergic neuromodulation we have confirmed the expression of the five *Drosophila* serotonin receptors in a subset of experimentally tractable cells in the lamina and used live imaging to determine their physiological effects. Our data are generally consistent with two previous transcriptomic studies by Davis et al. [70] and Konstantinides et al. [71] and indicate that L2 cells express 5-HT2B, T1 cells express both 5-HT1A and 5-HT1B, and L1 does not express detectable levels of any serotonin receptors. Our data are also consistent with the expression of 5-HT1A in C2 cells and 5-HT7 in L5 cells. In S2 Fig we directly compare our current findings with those of Davis et al. [70] and Konstantinides et al. [71]. Although our RT-qPCR data support the possibility

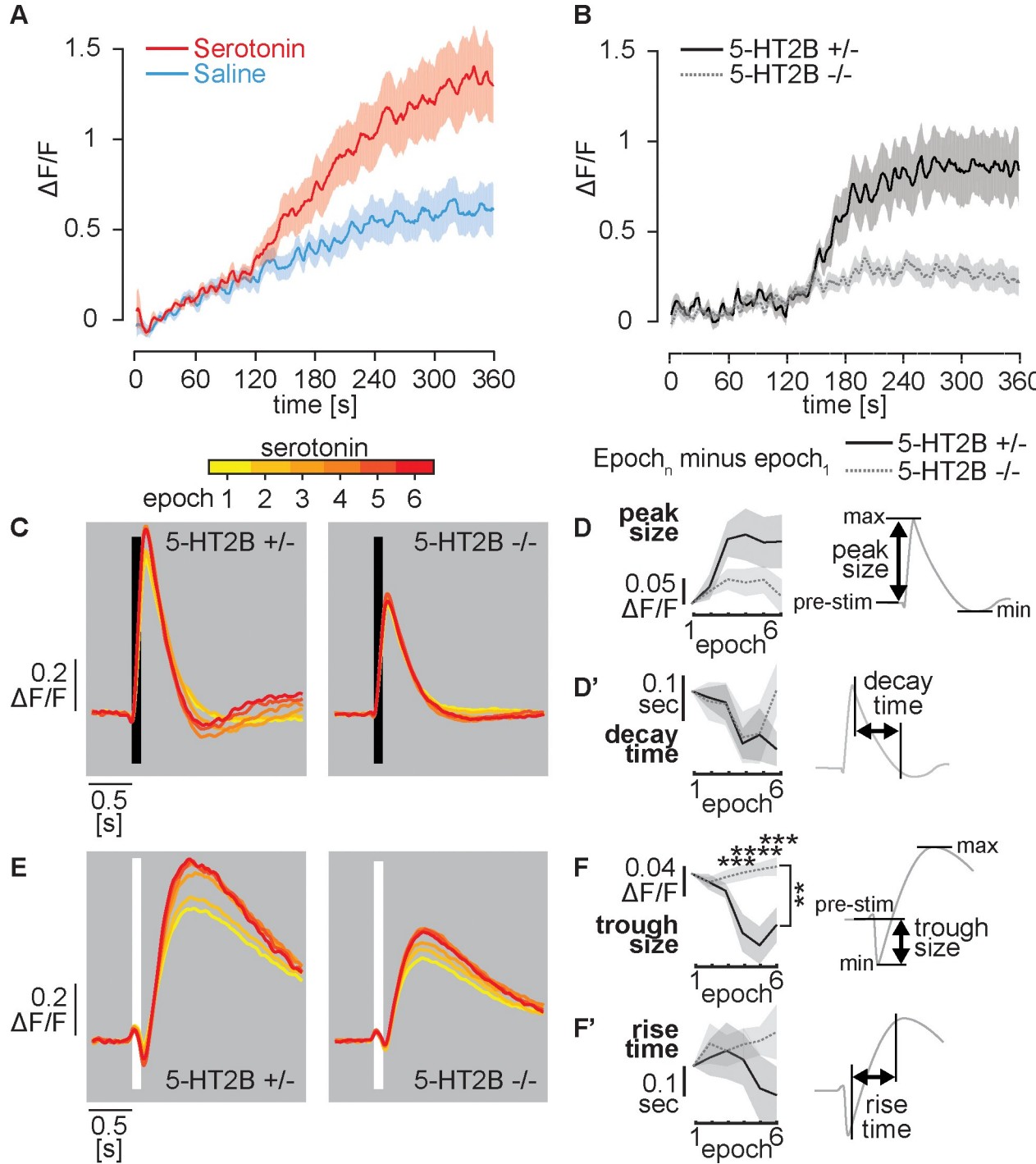

**Fig 6. 5-HT2B mediates the L2 neuron response to serotonin.** Perfusion with serotonin or saline alone was initiated ~105 s after baseline recording in saline as in Fig 5. (**A**) L2>GCaMP6f response of wild 5-HT2B +/+ flies to serotonin versus saline perfusion in the absence of TTX. (**B**) L2>GCaMP6f response of homozygous (5-HT2B -/-) and heterozygous (5-HT2B +/-) flies to serotonin perfusion. (**C-F**) Calcium transients following brief light or dark flashes in homozygous (5-HT2B -/-) and heterozygous (5-HT2B +/-) flies. Metrics and analysis are identical to those described in Fig 5. For (A), N = 14 serotonin and N = 20 saline exposed flies were tested. For (B-F), N = 13 5-HT2B +/- and N = 15 5-HT2B -/- flies, all receiving serotonin, were tested. Shaded areas show mean +/- SEM. Comparisons in (D, F) are two-way repeated measures ANOVA (brackets show interactions between time and genotype) and Sidak's multiple comparisons tests for indicated time points, p≤0.01**, p≤0.001***, p≤0.0001****.

that L2 neurons might also express 5-HT7, neither we nor others detect an enrichment of 5-HT7 using RNA-Seq [70].

The MCFO stochastic labeling approach used in Fig 1 is based on the assumption that MiMIC-driven GAL4 expression is transcriptionally linked to gene expression and so may be more likely to occur in cells where the gene is more highly expressed. For this reason, the MiMIC-based screening approach could be biased toward more highly expressed genes and may provide a less comprehensive picture of expression compared to sequencing approaches. Although Konstantinides et al. [71] report expression of 5-HT1A and two additional receptors (5-HT1B and 5-HT7) in C2 neurons, they report expression levels for 5-HT1A approximately 30 times higher than either 5-HT1B or 5-HT7. We did not detect 5-HT1B or 5-HT7 in centrifugal C2/C3 cells, and this may result from lower levels of expression in these cell types. Unlike Davis et al. [70] we did not detect expression of serotonin receptors in either photoreceptor cells or in lamina monopolar neuron L3 using MiMIC-T2A-GAL4>UAS-MCFO. In addition to low levels of expression, the inconsistencies between data sets could be due to biological variability in levels of gene expression or infidelity in the MiMIC-based approach, which may not perfectly reflect endogenous expression. Differences between our genomic data and that of others may also reflect variation in the methods used to isolate target cell populations, extract RNA, or create DNA libraries for sequencing. For example, TAPIN-seq involves successive rounds of immunoprecipitation, which can be affected by non-specific antibody binding to ambient RNA or off-target proteins. By contrast, our use of FACS can lead to contamination by other cell types due to incomplete dissociation of GFP-tagged cells or due to non-specific labeling by the driver lines.

Consistent with the predicted coupling of 5-HT2B to $G_q$, we found that L2 neurons respond to serotonin with a robust increase in basal calcium measured by GCaMP6f fluorescence (Fig 4B, Fig 6A, S9A Fig) and this effect was dramatically reduced in 5-HT2B -/- flies (Fig 6B). The simplest explanation for these effects is that 5-HT2B expressed in L2 regulates the increase in basal calcium. Specific knockdown of 5-HT2B in L2 neurons will be required to address this important issue. Since RNAi lines can be ineffective or yield off-target effects, additional genetic tools may be useful to perform cell type specific knock down or knock out experiments [109]. Additional experiments will also be needed to assess the potential developmental effects of the 5-HT2B mutant, and in particular how loss of 5-HT2B may have altered the baseline response of L2 to visual stimuli in the absence of exogenous serotonin. Finally, we cannot completely rule out the possibility that 5-HT7 could contribute to the regulation of at least some L2 cells (see S7B Fig). These limitations aside, our data indicate that 5-HT2B induces robust increases in basal levels of intracellular calcium in L2 neurons.

In L1 neurons, which do not express serotonin receptors, we unexpectedly observed a large basal calcium response to serotonin similar to that of L2 neurons (Fig 4C). The most likely explanation would seem to be activation of L1 neurons by input from cells that express serotonin receptors either in the optic lobes or perhaps the central brain. GABAergic C2 neurons [110] may express 5-HT1A (S2 Fig, S3D and S3E Fig) and are known to synapse onto L1 in M1 [36]. Likewise, L5 neurons may express the 5-HT7 receptor (Fig 1E, S2 Fig) and reciprocally synapse with L1 neurons in M1 and M5 [36]. L1 recordings were performed in the presence of TTX, which blocks the *Drosophila* sodium channel $DmNa_V$ formerly (*para*) [111, 112] and represents a standard method to reduce neuronal inputs [103,104]. While TTX is expected to inhibit most if not all action-potentials, graded potentials, which are common in the fly [113–115] could activate L1 if these inputs were not influenced by inhibition of $DmNa_V$. Alternatively, indirect serotonergic regulation of L1 neurons might occur through gap junctions previously shown to couple L1 and L2 neurons [28]. In addition, we and others [70,84] detect serotonin receptor expression in glia (S4 Fig), and it is possible that activation of the serotonin

receptors in lamina glia could indirectly influence L1, perhaps via regulation of extracellular ion concentrations [116,117].

The only prior report of an acute physiological response to serotonin in the *Drosophila* visual system described the modulation of potassium channels in photoreceptors [44]. However, the ex vivo preparations used for these experiments may have included other cells or cell fragments, and unlike a previous transcriptomic study [70], we did not detect serotonin receptor expression in photoreceptors. We speculate that, similar to L1, serotonin may indirectly regulate photoreceptors via inputs from other neurons such as L2 or lamina glia [27,44,118].

Serotonin is implicated in circadian rhythms [55,56] and both L1 and L2 terminals exhibit rhythmic daily size changes in the medulla that are mediated by both serotonin [59] and glia [119]. Tissue levels of serotonin levels also decrease in constant darkness [56], and T1 neurons have been shown to be involved in circadian entrainment via expression of *timeless2* [120]. It is tempting to speculate that the acute increase in basal calcium we observed in response to exogenous serotonin (Fig 4B and 4C) could be relevant to circadian changes in the function of L1 and L2. In future experiments, it will be interesting to examine more chronic effects of serotonin on the physiology of L2 and whether 5-HT2B is also involved in these phenomena.

In T1 neurons, we were unable to detect any acute changes in basal calcium or voltage in response to serotonin application (Fig 4D and S9C and S9D Fig) and other probes (e.g., for cAMP) may be necessary to detect the acute response of T1 to serotonin. However, it is also possible that activation of 5-HT1 receptors does not induce any acute physiological response, and that more chronic indices will be necessary to detect the potential effects of serotonin on T1 neurons. For now, the absence of a GCaMP6f response in T1 neurons serves as an important negative control and demonstrates the specificity of the responses observed in L1 and L2.

Based on the localization of the tagged 5-HT2B receptor, we suggest that serotonin acts on L2 terminals in the medulla neuropil. While we do not know the physiological effects of serotonin on T1, we find that a tagged version of 5-HT1A expressed via the 5-HT1A locus in T1 and other cells localized to a similar site in the medulla neuropil (Fig 3). The localization of 5-HT1A and 5-HT2B to nerve terminals suggests that they could regulate presynaptic neurotransmitter release. If so, future experiments may reveal that serotonergic inputs to L2 regulate the physiological response to visual stimuli in neurons that are post-synaptic to L2.

Coordinated regulation of multiple terminals by a single serotoninergic neuron may be facilitated by the extrasynaptic release of serotonin through volume transmission, the primary mode of signaling in most aminergic circuits within the mammalian CNS [48,121–124]. Serotonergic release sites appear to lack synaptic partners in the lamina of the blowfly *Calliphora* [125] and our data using sybGRASP to test whether serotonergic neurons synapse onto L2, T1, and L1 was negative (S8 Fig). Recent advancements such as the full adult fly brain EM dataset [126] combined with predictive neurotransmitter mapping [127] will be important resources to determine the precise relationship between serotonergic boutons and the nerve terminals within the medulla neuropil. Further studies will also be needed to define the function of putative serotonin autoreceptors which we and others detect in serotonergic neurons (S5 Fig, S6 Fig and Fig 3B) [56,89]. While the importance of serotonergic autoreceptors in mammalian circuits is well established [128–130], their function in the fly is not well described.

We observed serotonin-mediated increases in basal calcium in L2 neuron terminals, which correlated with more subtle modulation of visually induced calcium transients following serotonin application (Figs 5 and 6). It is possible that the modest changes in calcium transient dynamics represent a role for serotonin in potentiating the response of L2-dependent visual processing pathways [31]. However, it is difficult to rule out the possibility that the non-linear dynamics of GCaMP6f could influence and perhaps account for the differences we observe between groups. Further experiments using other reporters may be useful to confirm our

observations. If validated, our data may suggest a molecular-mechanism for previous observations made in larger insects including serotonin-induced changes field potentials in blowfly representing the output of lamina monopolar cells [53] and honeybee motion detection in the lobula [45].

We observed differences in the visual responses of 5-HT2B +/- and 5-HT2B -/- even before serotonin perfusion. This might result from the activation of 5-HT2B by endogenously released serotonin or developmental defects caused by loss of 5-HT2B. It is possible that such developmental defects could broadly disrupt the physiology of L2 neurons and thereby confound our interpretation of the cell's response to exogenous serotonin. Future experiments to inducibly block the activity of 5-HT2B and/or endogenous serotonin in the adult but allow wild type activity during development will be needed to address this possibility. While the serotonin concentration used in our experiments is comparable to other studies in *Drosophila* [89,99,100] it is difficult to mimic the physiological environment using bath applied neurotransmitters. Future experiments using optogenetic stimulation will be needed to address this issue.

The relationship between repeated visual stimuli and basal calcium levels will also require further investigation. During the initial period before exogenous serotonin perfusion, the basal calcium levels of flies exposed to repeated visual stimuli (Fig 6A) increased while those shown a constant, medium luminance screen (Fig 4B) did not appreciatively change. We speculate that the repeated visual stimuli may have induced the modest increase in basal calcium observed in Fig 6A as part of an adaptive response. It is possible that activation of 5-HT2B by endogenously released serotonin could be responsible for this effect.

The increase in basal calcium following serotonin perfusion could potentially affect previously identified properties of L2. These include surround inhibition mediated by the GABA-A channel Rdl [131]. If rising basal calcium is accompanied by a tonic membrane depolarization further from the Cl⁻ reversal potential, the driving force on GABA-A mediated Cl⁻ current would be expected to increase. If so, serotonin might change the spatial structure of surround inhibition as well as tip the balance between ON and OFF response components to favor ON stimulation. Cell-specific manipulation of *Rdl* and/or 5-HT2B might be used to test this hypothesis [109,132].

Studies in mammals have already begun to dissect the contributions of serotonergic tuning in multiple cells within individual circuits including the visual system [6,18,23,133,134] but the way in which this information is integrated remains poorly understood. The interactions between receptors expressed on L2 and other neurons in the fly visual system provide a new framework to dissect the mechanism by which multiplexed serotonergic inputs combine to regulate circuit function.

## Conclusion

We have demonstrated that a subset of cells in the lamina express serotonin receptors and respond to exogenous serotonin with a robust increase in basal calcium and a more modest change in their physiological response to visual stimuli. Further experiments will be needed to determine how these changes affect circuit output and processes that occur over longer time scales.

## Methods

### Fly husbandry and genetic lines

Flies were maintained on a standard cornmeal and molasses-based agar media with a 12:12 hour light/dark cycle at room temperature (22–25˚C). All fly strains used in this study are

listed in S3 Table. Serotonin receptor MiMIC-T2A-GAL4 lines described in [77] were a gift from Herman Dierick (Baylor College of Medicine), and include 5-HT1A-T2A-GAL4[MI01468], 5-HT1A-T2A-GAL4[MI01140], 5-HT1A-T2A-GAL4[MI04464], 5-HT1B-T2A-GAL4[MI05213], 5-HT2A-T2A-GAL4[MI0459], 5-HT2A-GAL4[MI03299], 5-HT2B-T2A-GAL4[MI06500], 5-HT2B-T2A-GAL4[MI5208], 5-HT2B-GAL4[MI7403], and 5-HT7-GAL4[MI00215]. L1- and T1-split-GAL4 lines [90], as well as unpublished LexA lines for L1 and T1, were generously provided by Aljoscha Nern and Gerry Rubin (HHMI/Janelia Research Campus). S.L. Zipursky (UCLA) generously provided L2-split-GAL4 and L2-LexA (RRID:BDSC_52510). Yi Rao (Peking University) generously shared 5-HT2B-KO-GAL4 SII (5-HT2B mutant) [93], 5-HT1A::sfGFP [92], and UAS-5-HT2B::sfGFP [93].

SerT-GAL4 (RRID:BDSC_38764), GAD1 Trojan LexA (RRID:BDSC_60324) and ChAT Trojan LexA (RRID:BDSC_60319) were obtained from Bloomington *Drosophila* Stock Center (BDSC) at Indiana University (Bloomington, IN, USA). Additional reporters from BDSC include: UAS-mCD8::GFP (RRID:BDSC_5137), UAS-MCFO-1 (RRID:BDSC_64085), UAS-GCaMP6f (RRID:BDSC_42747), UAS-ArcLight (RRID:BDSC_51056), UAS-mCD8::RFP, LexAop-mCD8::GFP (RRID:BDSC_32229), and UAS-nSyb::GFP1-10, LexAop-CD4:GFP11 (RRID:BDSC_64314; provided through S.L. Zipursky (UCLA)).

## Immunofluorescent labeling and imaging

Flies were dissected 5–10 days after eclosion, and equal numbers of males and females were used for all experiments unless otherwise noted. Brains were dissected in ice-cold PBS (Alfa Aesar, Cat#J62036, Tewksbury, MA), then fixed in 4% paraformaldehyde (FisherScientific, Cat#50-980-493, Waltham, MA) in PBS with 0.3% Triton X-100 (Millipore Sigma, Cat#X100, Burlington, MA) (PBST) for one hour at room temperature. After fixation, brains were washed three times with PBST for 10 minutes, then blocked for 30 minutes in PBST containing 0.5% normal goat serum (NGS) (Cayman Chemical, Cat#10006577, Ann Arbor, MA). Antibodies were diluted in 0.5% NGS/PBST. Primary antibodies were incubated with the tissue overnight at 4˚C. The next day, the brains were washed three times with PBST for 10 minutes, then incubated with secondary antibodies for 2 hours in the dark at room temperature. Brains were washed three times with PBST for 10 minutes before mounting.

For frontal mounting, brains were washed with 60% and 80% glycerol (Millipore Sigma, Cat#G5516) and mounted in Fluoromount-G (SouthernBiotech, Cat#0100–01, Birmingham, AL). For dorsal-ventral mounting, brains were fixed in 2% PFA/PBST overnight. The next day, brains were washed three times with PBST. Brains were dehydrated with a series of 10 min ethanol baths of increasing concentrations (30%, 50%, 75%, 95%, 100%, 100%, and 100%). Brains were then transferred to 100% xylene before mounting in DPX (FisherScientific, Cat#50-980-370).

Serotonin immunolabeling was performed with 1:25 rat anti-serotonin (Millipore Sigma, Cat#MAB352, RRID:AB_11213564), 1:1000 rabbit anti-serotonin (ImmunoStar, Cat#20080, Hudson, WI, RRID:AB_572263) or 1:1000 goat anti-serotonin (ImmunoStar, Cat#20079, RRID:AB_572262). Where noted, GFP was labeled with 1:250 mouse anti-GFP (Sigma-Aldrich, Cat#G6539, RRID:AB_259941; or, ThermoFisher, Waltham, MA, Cat#A-11120, RRID:AB_221568). Secondary antibodies were used at 1:400 and include: Donkey anti-mouse Alexa Fluor 488, Donkey anti-Rabbit Alexa Fluor 594 or Alexa Fluor Donkey anti-rat 647 (Jackson ImmunoResearch Laboratories, Westgrove, PA, Cat#715-545-151, # 711-585-152, # 712-605-153) or Alexa Fluor 555 (Life Technologies, ThermoFisher, Cat#A-21428).

When serotonin receptor MiMIC-GAL4 lines were combined with ChAT or GAD1 MiMI-C-LexA (S3 Fig), brains were processed and imaged as described in Sizemore and Dacks 2016 [135]. MultiColor FlpOut (MCFO-1) sparse labeling was induced by heat activation at 37˚C for

10–15 minutes at least 2 days prior to dissection as described [82]. Although MCFO can be used for lineage tracing [82], we induced MCFO in adult flies, when visual system neurons are post-mitotic, and MCFO labeling does not represent clonal events. Primary antibodies included 1:300 rabbit anti-HA (Cell Signaling Technology, Cat#3724, Danvers, MA, RRID:AB_1549585), 1:150 rat anti-FLAG (Novus, Littleton, CA, Cat#NBP1-06712, RRID:AB_1625982), and 1:400 mouse anti-V5::Dylight-550 (Bio-Rad, Hercules, CA, Cat#MCA1360D550GA, RRID:AB_2687 576). The rat anti-N-Cadherin (DN-Ex#8) and mouse anti-repo (8D12 concentrate) was obtained from the Developmental Studies Hybridoma Bank, created by the NICHD of the NIH and maintained at The University of Iowa, Department of Biology, Iowa City, IA 52242. Secondary antibodies used for MCFO are listed above. N-Synaptobrevin GFP Reconstitution Across Synaptic Partners (sybGRASP) flies [94] were dissected, fixed and immunolabeled as described above, without KCl induction. The tissue was labeled with mouse antiserum specific to reconstituted GFP (1:250; Sigma-Aldrich, Cat#G6539, RRID:AB_259941) [136] and either anti-serotonin (antibodies listed above).

Imaging was performed with a Zeiss LSM 880 Confocal with Airyscan (Zeiss, Oberkochen, Germany) using a 40x water or 63x oil immersion objective. Post-hoc processing of images was done with Fiji [137] or Adobe Photoshop (Adobe, San Jose, CA).

## FACs and RNA extraction

L2 and L1 neurons were labeled using split-GAL4 drivers combined with UAS-mCD8::GFP (RRID:BDSC_5137). For RNA-Seq in Fig 2, N = 3 T1-LexA samples were tested. For RT-qPCR in S7 Fig, we included N = 3 T1-split-GAL4 samples and N = 3 T1-LexA samples. Brains were dissected on the day of eclosion and optic lobes were dissociated according to previously published methods [138]. The dissociated optic lobe cells were separated by fluorescence-activated cell sorting (FACS) into GFP-positive and GFP-negative isolates using a BD FACS Aria II high-speed cell sorter in collaboration with the UCLA Jonsson Comprehensive Cancer Center (JCCC) and Center for AIDS Research Flow Cytometry Core Facility (http://cyto.mednet.ucla.edu/home.html). For FACS, each experiment was performed with 18–40 brains, and yielded between 1,700–7,800 GFP$^+$ cells. RNA was extracted from isolated cells with ARCTURUS® PicoPure® RNA Isolation Kit (ThermoFisher, KIT0204) or RNeasy Plus Micro Kit (QIAGEN, 74034).

## RT-qPCR

RNA extracted from FACS isolates was reverse transcribed using SuperScript III (Invitrogen, ThermoFisher, Cat#18080093). RT-qPCR was performed for receptor cDNA using validated primers (S4 Table) and SYBR Green Power PCR Mix (Applied Biosystems, ThermoFisher) on an iQ5 real-time qPCR detection system (Bio-Rad). Primers were designed using Primer-Blast (https://www.ncbi.nlm.nih.gov/tools/primer-blast/) or were from the DGRC FlyPrimerBank [139]; oligonucleotides were obtained from Integrated DNA Technologies (Coralville, Iowa). Primer pairs were validated to amplify a single product, verified by a single melting temperature and single band on an electrophoresis gel. The efficiency for each primer pair was between 85–115%. Comparisons between GFP$^+$ and GFP$^-$ samples were calculated as enrichment (i.e., fold change) using the comparative CT method [91]. A zero value was imputed for samples with no amplification (i.e., no CT value). Raw CT values are shown in S2 Table.

## RNA-Seq

RNA-Seq was performed using a SMART-Seq protocol adapted from [138,140,141]. Libraries were constructed using the SMART-seq v4 Ultra Low-input RNA sequencing kit with Nextera XT (Takara Bio). Paired-end sequencing was conducted by the UCLA genomic core facility

(https://www.semel.ucla.edu/ungc/services). After demultiplexing, we obtained between 39–270 (average 105) million reads per sample. Quality control was performed on base qualities and nucleotide composition of sequences. Alignment to the *Drosophila melanogaster* genome (BDGP6) was performed using the STAR spliced read aligner [142] with default parameters. Additional QC was performed after the alignment to examine the following: level of mismatch rate, mapping rate to the whole genome, repeats, chromosomes, and key transcriptomic regions (exons, introns, UTRs, genes). Between 75–85% of the reads mapped uniquely to the fly genome. Total counts of read fragments aligned to candidate gene regions within the reference gene annotation were derived using HTSeq program and used as a basis for the quantification of gene expression. Only uniquely mapped reads were used for subsequent analyses. Following alignment and read quantification, we performed quality control using a variety of indices, including consistency of replicates and average gene coverage. For Fig 2, L2 samples were run in two separate sequencing runs and we did not perform corrections for any potential batch effects. Data is shown as Transcripts Per Million (TPMs). The RNA-Seq data (raw and processed files) are available on GEO at https://www.ncbi.nlm.nih.gov/geo/query/acc.cgi?acc=GSE154085.

## Live cell imaging

Calcium imaging was performed as previously described [132,143,144]. Briefly, flies were anesthetized at 4˚C and placed into a chemically etched metal shim within a larger custom-built fly holder. The fly holder was based on a previously described design [145]. The head capsule and the thorax were glued to the metal shim using UV-curable glue (www.esslinger.com). The legs, proboscis and antennae were immobilized using beeswax applied with a heated metal probe (Waxelectric-1, Renfert). The head capsule was immersed in insect saline (103 mM NaCl, 3 mM KCl, 1.5mM CaCl2, 4 mM MgCl2, 26 mM NaHCO3, 1 mM NaH2PO4, 10 mM trehalose, 10 mM glucose, 5 mM TES, 2 mM sucrose) [146]. A small window on the right rear head capsule was opened using sharp forceps (Dumont, #5SF). Muscles and fat covering the optic lobe were cleared before placing the fly under the 2-photon microscope (VIVO, 3i: Intelligent Imaging Innovations, Denver, CO). Neurons expressing GCaMP6f were imaged at 920-nm using a Ti:Sapphire Laser (Chameleon Vision, Coherent). Images were acquired at 10–20 frames/s for Fig 4 and 25–30 frames/s for Figs 5 and 6 live imaging. Only female flies were used for live imaging experiments.

A custom-built gravity perfusion system was used for bath application of either serotonin or saline control to the fly's exposed optic lobe for Fig 4. For Fig 4 and S9 Fig, the tissue was first perfused with insect saline containing 1μm tetrodotoxin citrate (TTX) (Alomone Labs, Jerusalem, Israel, Cat#T-550) for at least 5 minutes at 2 mL/min, prior to each recording. TTX remained present throughout the experiment. To examine the effects of serotonin on calcium levels, baseline GCaMP6f fluorescence was recorded for one minute before switching to the second input containing either 100 μM serotonin hydrochloride (Sigma Aldrich, Cat# H9523) or saline alone for an additional five minutes of recording. The concentration of serotonin used is comparable to recent physiological studies applying exogenous serotonin to the *Drosophila* brain [89,99,100]. Due to perfusion tubing length and dead volume, the perfusion switch took approximately 105 s to reach the tissue. Fig 4 does not show the first minute of the recording, so the solution switch occurs at approximately 45 s on the x axis. For more precise control of perfusion solutions, we used a programmable valve controller (VC-6, Warner Instruments, Hamden, CT) for Figs 5 and 6 visual experiments (see details below).

## Visual stimulus experiments

Visual stimuli were shown using an arena composed of 48 eight by eight-pixel LED panels, at 470 nm (Adafruit, NY, NY). The panels were assembled into a curved display that extends

216˚ along the azimuth and ±35˚ in elevation. Each pixel subtended an angle of 2.2˚ on the retina at the equatorial axis. To prevent spurious excitation of the imaging photomultiplier tubes, three layers of blue filter (Rosco no. 59 Indigo) were placed over the LED display.

Each stimulus consisted of a brief increment (light flash) or decrement (dark flash) of the entire display for 100 ms, before returning to a mid-intensity brightness for 4.9 s. Images were acquired at 25–30 frames/s for Figs 5 and 6 visual stimulation experiments. Stimuli were presented in sets of six bright and six dark flashes randomly shuffled for each minute of the experiment. Responses were then pooled for each minute. During the first minute ("epoch 1" in Figs 5 and 6), and prior to imaging, the tissue was perfused with saline for a baseline recording. At the end of the first minute, a valve controller (VC-6, Warner Instruments, Hamden, CT) activated by a TTL signal switched the perfusion to either saline with 100 μM serotonin or saline alone; imaging then continued for an additional five minutes, for a total of one baseline set and five post-switch sets of stimuli. The perfusion switch took approximately 45 s to reach the tissue using the programmable valve system.

## Analysis

The analysis code and the live-imaging data has been deposited at https://osf.io/39j4m/ (DOI: 10.17605/OSF.IO/39J4M). Calcium imaging data were analyzed with Matlab R2017a (Mathworks, Natick, MA). Post hoc, recordings were corrected for movement of the brain within the imaging plane using a custom algorithm [147]. Regions of interest (ROIs) were found semi-automatically for data in Figs 4–6: first, the median intensity of all pixels across all image frames was found; this value was used as a threshold and all pixels with mean intensity below the threshold, typically within the image background, were discarded. The 1-D time series of intensity for each remaining pixel was then extracted. K-means clustering was used to identify pixels with similar activity over the course of the experiment: three clusters were identified and the cluster with the highest number of pixels was retained. This reliably identified the pixels within active neurons in the imaging data and aided in identifying preparations with out-of-plane movement, which were discarded.

For basal calcium experiments (Fig 4), the remaining cluster was used as a single ROI and the mean intensity within the ROI was found for each image frame to produce a single time-series for the entire experiment. For visual response experiments (Figs 5 and 6), pixels within the remaining cluster were automatically divided into groups corresponding to individual L2 terminals using a watershed transform. The mean intensity within each ROI was found for each image frame to produce a single time-series for the entire experiment, and the time-series for all terminal ROIs within an individual animal were then averaged. For Fig 6, ROIs of L2 terminals were first identified automatically, as above, then manually selected individually according to layer position because the 5-HT2B GAL4 SII mutant line labeled other cells in addition to L2 neurons.

Approximately half of the bath application recordings showed oscillations in activity due to slow, periodic movement of the brain at around 0.04 Hz; we applied a second-order notch filter at this frequency with a bandwidth of 0.005 Hz to remove these oscillations. For the bath application experiments (Fig 4), we plotted $\Delta F/F$, defined as $(F_t-F_0)/F_0$, where $F_t$ is the mean fluorescence in the ROI at the indicated time and $F_0$ is the mean value of $F_t$ during 60 seconds of baseline activity at the beginning of the experiment and prior to the change in perfusion. For the visual stimulus experiments (Figs 5 and 6), we again plotted $\Delta F/F$, defined as $(F_t-F_0)/F_0$, where $F_t$ is the mean fluorescence across all individual terminal ROIs at the indicated time and $F_0$ is the mean of 30 seconds of non-consecutive baseline activity between stimulus presentations during epoch 1 at the beginning of the experiment and prior to the change in perfusion

(Fig 5A). For the stimulus response plots (Figs 5B, 5D, 6C and 6E), we found the average $\Delta F/F$ time-series within each epoch for each fly after subtracting the average pre-stimulus baseline activity level (0.5 s preceding each flash stimulus) from each time-series, so that all responses started aligned at 0 $\Delta F/F$. For further analysis (Figs 5C, 5E, 6D and 6F), we calculated the changes in response amplitude across epochs, defined for the dark stimulus presentation as the difference between the pre-stimulus baseline and the maximum post-stimulus $\Delta F/F$ value. For each $epoch_n$, we subtracted the value of the responses during epoch 1 at the beginning of the experiment and prior to the change in perfusion (indicated in the Figs 5 and 6 as "$Epoch_n$ minus $epoch_1$"), in order to find the change in amplitude relative to epoch 1. We followed a similar procedure for the secondary calcium responses (S10-S11) the decay (or rise) time (Figs 5C', 5E', 6D' and 6F'), defined as the length of time between 10% and 90% of the post-stimulus peak-to-peak difference in $\Delta F/F$, and the exponential time constant k defined by $\Delta F/F(t) = ce^{kt}$.

In the text we use "basal calcium signal" to indicate $\Delta F/F$ readings that follow a relatively slow time course and do not appear to be in response to visual stimuli (see Figs 4B,4C, 5A, 6A and 6B). The $\Delta F/F$ readings that occur during the first sixty seconds of the Figs 5 and 6 experiments ("Epoch 1") prior to changing the perfusion solution are indicated as "baseline" values. The Epoch 1 "baseline" includes values for both the *basal calcium level* and the magnitude of the *calcium transients* seen in response to visual stimuli.

To examine the changes in fluorescence representing changes in basal calcium in visual experiments shown in Fig 6A and 6B responses to the visual stimuli were removed using a series of second-order notch filters at 0.19–0.21 Hz and 0.38–0.42 Hz.

### Statistical tests

For (Fig 5C–5C" and 5E–5E") and (Fig 6D–6D" and 6E–6E") comparisons are two-way repeated measure ANOVA (brackets show interactions between time and genotype) and Sidak's multiple comparisons tests, $p \leq 0.05$ *, $p \leq 0.01$**, $p \leq 0.001$***, $p \leq 0.0001$****. These tests were performed using Graphpad Prism Software (San Diego, CA). Differences in basal calcium shown in Fig 6A and 6B were calculated by two-tailed Wilcoxon rank sum tests in Matlab R2017a.

### Replicates

Each biological replicate (N) represents one fly, except for RT-qPCR and RNA-Seq (Fig 2, and S7E Fig) where each biological replicate was pooled from 18+ flies. RT-qPCR experiments include 3 technical replicates, which are averaged to represent a single biological replicate. Animals from at least 3 crosses were used for each experiment. Data for each experiment was collected over 2–6 months in at least 3 experiments. No outliers were removed from any data set. Live imaging recordings with too much movement were excluded and not analyzed.

### Supporting information

**S1 Fig. Serotonin receptors and serotonergic projections in the optic lobe.** (**A-H**) Serotonin receptor MiMIC-T2A-GAL4 lines were crossed to UAS-mCD8::RFP (visualized here in green) to identify patterns of expression in the optic lobe. (**A**) A schematic of the optic lobe neuropils including the lamina (la), medulla (me), lobula (lo) and lobula plate (lp) with the neuropil in grey and the cortex containing neuronal cell bodies in white. (**B-F**) Neuropil is labeled by anti-N-Cadherin staining (blue) to provide anatomical reference for labeled cells (green) representing 5-HT1A (**B**), 5-HT1B (**C**), 5-HT2A (**D**), 5-HT2B (**E**), and 5-HT7 (**F**). MiMIC-T2A-GAL4 line labeled projections were visible in all optic lobe neuropils including the lamina neuropil,

which is enlarged in (B'), (C'), (E') and (F'). N = 4–8 and scale bars are 20 µm.
(TIF)

**S2 Fig. Data sets reporting evidence of serotonin receptor expression in optic lobe neurons.** The current study includes MiMIC-T2A-GAL4>MCFO for identification based on morphology (green) and FACS-SMART-Seq of L2 and T1 neurons (blue). Davis et al. 2020 [70] employed TAPIN-Seq and reported probability of expression (GSE116969, Table 7B) for each cell type. Serotonin receptor expression with a p>0.75 are shown (purple). Konstantinides et al. 2018 [71] used FACS-SMART-Seq for T1, Mi1, C2 and C3 cells (GSE103772). Serotonin receptors with counts greater than 1,000 in at least two replicates are shown (orange).
(TIF)

**S3 Fig. Serotonin receptor MiMIC-T2A-GAL4 lines potentially label L2, C2, TMY3, and Mi1 cells.** (**A**) 5-HT2B-MiMIC-T2A-GAL4>UAS-RFP (green) was combined with ChAT-MiMIC-LexA>LexAop-GFP (magenta). Co-labeling was observed in cell bodies in the lamina cortex, shown in insets (arrowheads). (**B-C**) 5-HT7-MiMIC-T2A-GAL4 labeled cells with a morphology similar to lamina monopolar cell 5 (L5). (**D**) 5-HT1A-MiMIC-T2A-GAL4>MCFO labeled C2-like cells in the lamina neuropil. (**E**) Colocalization was observed between 5-HT1A MiMIC-T2A-GAL4 and GAD1 MiMIC-T2A-LexA in the lamina neuropil and cell bodies adjacent to the lobula plate (arrowhead, E). (**F-G**) 5-HT7-MiMIC-GAL4>MCFO (F) and 5-HT1A-MiMIC-T2A-GAL4>MCFO (G) labeled cells with morphology similar to TmY3. (**H**) 5-HT7-MiMIC-GAL4>MCFO also labeled cells that resembled Mi1. For co-labeling in (A) and (E), N = 4–8 brains per condition. For MCFO, L5-like cells in (B-C) were observed in 7/13 brains, C2 cells in (D) were observed in 3/31 brains, TMY3 cells in (F) were observed in 4/13 brains, TmY3 cells in (G) were observed in 5/31 brains and Mi1 cells in (H) were observed in 6/13 brains. Scale bars are 20 µm for (A, D-H) and 10 µm for (B-C).
(TIF)

**S4 Fig. 5-HT2A labeling in lamina cortex may represent glia cells.** (**A**) 5-HT2A-GAL4>MCFO epitopes V5 (green) and HA (magenta) label unidentified cells confined the distal lamina cortex. (**B**) 5-HT2A-T2A-GAL4>UAS-mCD8::GFP (green) labels cells in the lamina cortex in close proximity to nuclei labeled with repo antibody (magenta). Neuropil is labeled by anti-N-Cadherin staining (blue) to provide anatomical reference. N = 18 brains for (A) and N = 4 brains from (B). Scale bars are 20 um.
(TIF)

**S5 Fig. Serotonin receptor 5-HT1B co-labels with serotonin immunoreactive sites in optic lobe and cell bodies in the central brains.** Anti-serotonin immunolabeling (magenta) was used to identify serotonergic cells and projections in (A-F). (**A**) Serotonin immunoreactive sites (magenta) were visible in the optic lobe neuropils: lamina (la), medulla (me), lobula (lo) and lobula plate (lp). (**B**) A schematic of the optic lobe and its major neuropils. (**A'-A" and C-C"**) 5-HT1B-MiMIC-T2A-GAL4>UAS-mCD8::GFP labeled cells throughout the optic lobe with close apposition to serotonergic boutons. A neuron in serotonergic cell cluster LP2 is also labeled by 5-HT1B driven GFP (arrowhead, A-A"). (D) A schematic of the fly brain with dashed lines showing the approximate anatomical locations for (C) and (E). (E-E") Serotonin receptor MiMIC-T2A-GAL4 lines were crossed to UAS-MCFO-1 to label individual cells. Using 5-HT1B-MiMIC-T2A-GAL4>MCFO (green), we observed co-labeling between MCFO-labeled cells and serotonergic boutons (magenta) processes in the inner medulla (iM), medulla layer 4 (M4), and lobula (lo). (F-F") Anti-serotonin immunolabeling (magenta) co-labeled with 5-HT1B-MiMIC-T2A-GAL4>UAS-mCD8::GFP labeled cell bodies in the central brain. 5-HT1B-labeled Kenyon cells (KC) are labeled for anatomical reference in (F"). (G) The

approximate anatomical location for images in (F-F") are shown in the boundaries of the dashed line. Serotonin co-labeling was performed N = 5 for 5-HT1B>GFP (A-A", C-C", and F-F") and N = 6 brains for 5-HT1B>MCFO (E-E"). Scale bars are 20 μm.
(TIF)

**S6 Fig. Serotonin receptor 5-HT1A co-labels with serotonin immunoreactive sites in optic lobe and cell bodies in the central brain.** (**A**) Schematic of the optic lobe neuropils—lamina (la), medulla (me), lobula (lo) and lobula plate (lp)—and serotonergic PLP cells. (**B-B"**) 5-HT1A-MiMIC-T2A-GAL4 driving UAS-mCD8::GFP (green) was co-stained with anti-serotonin immunolabeling (magenta) to map potential autoreceptors to specific cell clusters in the central brain. PLP neurons co-labeling for 5-HT1A labeling and anti-serotonin immunolabeling are indicated by the arrowhead. (**C-E**) Anterior to posterior images taken in the same brain show several serotonergic cell clusters expressing 5-HT1A (labeled at arrowheads). 5-HT1A-labeled mushroom body (MB) and kenyon cells (KC) are labeled for anatomical reference in (**C** and **E**). (**F**) A cartoon of the fly brain with dashed lines to indicate the approximate anatomical location for (C-E). Scale bars are 20 μm and N = 6.
(TIF)

**S7 Fig. RT-qPCR shows L2 neurons express 5-HT2B and T1 neurons express both 5-HT1A and 5-HT1B serotonin receptors.** Enrichment (i.e., fold change) was calculated for cDNA from GFP-labeled cell isolates relative to pooled, unlabeled optic lobe cell isolates using the comparative CT method. The dotted line indicates y = 1. Values >1 signify that the transcript is more abundant (enriched) in GFP-labeled cells, while values <1 signify that these transcripts are less abundant in GFP-labeled cells A zero value was imputed for samples with no detectable transcript amplification (i.e., no CT value). (**A**) RT-qPCR performed on cDNA from isolated T1 neurons expressing GFP showed enrichment for serotonin receptors 5-HT1A and 5-HT1B relative to other GFP-negative cells from the optic lobe. (**B**) FACS isolates from L2 cells showed enrichment of 5-HT2B and 5-HT7 in RT-qPCR. (**C**) L1 RT-qPCR enrichment was not detectable for any serotonin receptors. RT-qPCR error bars represent mean±SEM. N = 3–6 biological replicates pooled from 18–40 brains per replicate.
(TIF)

**S8 Fig. Serotonergic neurons do not show sybGRASP signal with postsynaptic T1, L2 or L1 neurons in the medulla.** SybGRASP was used to probe whether serotonergic neurons make synaptic contacts onto L2, T1 or L1 neurons. L2 neurons are known to synapse onto T1 projections in the medulla and we used this connection as a positive control. (**A-A'**) SybGRASP was observed with L2 split-GAL4 presynaptic to T1-LexA in M2. The dashed inset in (**A**) is shown in (**A'**). (**B-E**) A SerT-GAL4 driver was used to express the pre-synaptic portion of GFP in serotonergic neurons and LexA drivers were used to express the postsynaptic portion of GFP in L2 (**B-C**) L1 (**D**) or T1 (**E**) as indicated. No sybGRASP signal was detected in the medulla when SerT was presynaptic to L2 (**B**) however, occasional sparse GFP puncta (arrowhead) were visible in the lamina (**C**). When SerT-GAL4 neurons were presynaptic to L1 (**D**) or T1 (**E**) neurons, we did not detect a sybGRASP signal in either the lamina or medulla. All tissue was labeled with primary antibodies to both 5-HT (magenta) and GFP (green). N = 7–10 brains. Scale bars are 15 μm (**A**, **B**, **D-E**); 5 μm (**A"** and **C-C"**).
(TIF)

**S9 Fig. Individual traces for serotonin bath application experiments.** (**A-D**) Individual traces representing all experiments (Fig 4) for serotonin or saline controls with L2, L1 or T1 split-GAL4>GCaMP6f or T1 split-GAL4>ArcLight. For all experiments, the first 60 s of the recording is not shown; traces represent data recorded following a switch to saline with

serotonin or saline alone. The length of time for the switch to complete was estimated to be 105 s, but the first 60 s of the recording are not shown so the switch occurs at 45 s on the x axis in A-D. Saline controls are gray, serotonin exposed preps are colored, and the dark line represents the mean. (**A**) L2>GCaMP6f experiments, along with (**B**) L1>GCaMP6f, show an increase in calcium following serotonin application as compared to saline controls (L2, p = 00095; L1, p = 0.02). (**C, D**) T1 cells show no significant change with either GCaMP6f (**C**) or ArcLight (**D**) relative to saline (p>0.05) For bath application experiments (**A-D**), N = 4–8 individual flies.
(TIF)

**S10 Fig. Additional Analysis of visually induced calcium transients in Fig 5.** Visually induced calcium transients were recorded in flies expressing L2-split-GAL4>GCaMP6f as in Fig 5. Bath application with saline followed by saline (blue) or serotonin (red) at 105 s. Analysis of visual stimuli response. For plotting each variable, the average value for the epoch 1 baseline was subtracted; epoch 1 is therefore always set to 0. (**A**) The change in the secondary response (the magnitude of the calcium decrease following the initial peak). (**B**) Change in the exponential time constant k (units 1/sec) relative to epoch 1 (defined by $\Delta F/F(t) = ce^{kt}$) for the time between 10% and 90% of the post-stimulus peak-to-peak difference in $\Delta F/F$. (**C**) Change in the secondary calcium response (the magnitude of the calcium increase following the initial trough). (**D**) Change in the exponential time constant relative to epoch 1. N = 14 and N = 20 individual flies perfused with serotonin or saline respectively. Shaded areas show mean +/- SEM.
(TIF)

**S11 Fig. Additional Analysis of visually induced calcium transients in Fig 6.** Visually induced calcium transients were recorded in 5-HT2B +/- or 5-HT2B -/- flies as described in Fig 6. (**A**) The change in the secondary response (the magnitude of the calcium decrease following the initial peak). (**B**) Change in the exponential time constant k (units 1/sec) relative to epoch 1 (defined by $\Delta F/F(t) = ce^{kt}$) for the time between 10% and 90% of the post-stimulus peak-to-peak difference in $\Delta F/F$. (**C**) Change in the secondary calcium response (the magnitude of the calcium increase following the initial trough). (**D**) Change in the exponential time constant relative to epoch 1. N = 13 5-HT2B +/- and N = 15 5-HT2B -/- flies, all receiving serotonin, were tested. Shaded areas show mean +/- SEM. Comparisons in are two-way repeated measures ANOVA (brackets show interaction of time and genotype) and Sidak's multiple comparisons tests for indicated time points, p≤0.05*.
(TIF)

**S1 Table. RNA-Seq Serotonin Receptor TPMs, averages and standard deviations.**
(PDF)

**S2 Table. RT-qPCR Threshold Cycle (CT) measurements and calculated enrichment for FACS-isolated T1, L2, and L1 samples as shown in S7 Fig.** Enrichment (i.e., fold change) was calculated for cDNA from GFP-labeled cell isolates relative to pooled, unlabeled optic lobe cell isolates using the comparative CT method.
(PDF)

**S3 Table. Fly strains used in this study.**
(PDF)

**S4 Table. RT-qPCR primer sequences and mRNA (cDNA) target information.**
(PDF)

## Acknowledgments

We thank members of Larry Zipursky's lab (UCLA) including Elizabeth Zuniga Sanchez, Liming Tan, and Juyoun Yoo for advice on FACS, RNA extraction, RT-qPCR and SMART-Seq. Thank you to Deanne Umbay (Brown University) and Nikoo Dalili (UCLA) for their assistance. We thank members of the Frye and Krantz labs for helpful discussions, Aljoscha Nern and Gerry Rubin (HHMI/Janelia Research Campus) and Herman Dierick (Baylor) for generously supplying fly lines.

## Author Contributions

**Conceptualization:** Mark A. Frye, David E. Krantz.

**Data curation:** Maureen M. Sampson, Ben J. Hardcastle, Shivan L. Bonanno.

**Formal analysis:** Maureen M. Sampson, Katherine M. Myers Gschweng, Ben J. Hardcastle, Shivan L. Bonanno, Fuying Gao.

**Funding acquisition:** Maureen M. Sampson, Tyler R. Sizemore, Andrew M. Dacks, Mark A. Frye, David E. Krantz.

**Investigation:** Maureen M. Sampson, Katherine M. Myers Gschweng, Ben J. Hardcastle, Shivan L. Bonanno, Tyler R. Sizemore, Fuying Gao.

**Methodology:** Maureen M. Sampson, Katherine M. Myers Gschweng, Ben J. Hardcastle, Shivan L. Bonanno, Mark A. Frye.

**Project administration:** Mark A. Frye, David E. Krantz.

**Resources:** Andrew M. Dacks, David E. Krantz.

**Software:** Ben J. Hardcastle.

**Supervision:** Andrew M. Dacks, Mark A. Frye, David E. Krantz.

**Validation:** Maureen M. Sampson, Katherine M. Myers Gschweng, Ben J. Hardcastle, Rebecca C. Arnold.

**Visualization:** Katherine M. Myers Gschweng, Tyler R. Sizemore.

**Writing – original draft:** Maureen M. Sampson, Katherine M. Myers Gschweng, David E. Krantz.

**Writing – review & editing:** Maureen M. Sampson, Katherine M. Myers Gschweng, Ben J. Hardcastle, Shivan L. Bonanno, Tyler R. Sizemore, Andrew M. Dacks, Mark A. Frye, David E. Krantz.

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
