## [Decision Letter · Decision Letter 0]

8 Jun 2020

Dear David,

Thank you very much for submitting your Research Article entitled 'Serotonergic modulation of visual neurons in Drosophila melanogaster' to PLOS Genetics. Your manuscript was fully evaluated at the editorial level and by three independent peer reviewers. The reviewers were very positive about the significance of the problem and about the work presented. However, they still had a number of issues and you will see a very long list of problems to address. These aspects of the manuscript should be improved in order to support its publication in PLOS Genetics.

One of the reviewers could not find the data availability statement in the methods (e.g. Calcium imaging or sequence data). Providing this is a PLOS Genetics policy, so please make sure that this is included with the revision.

We therefore ask you to modify the manuscript according to the review recommendations before we can consider your manuscript for acceptance. Your revisions should address the specific points made by each reviewer.

[LINK]

Yours sincerely,

Claude

Claude Desplan

Associate Editor

PLOS Genetics

Gregory P. Copenhaver

Editor-in-Chief

PLOS Genetics

Reviewer's Responses to Questions

**Comments to the Authors:**

Reviewer #1: In this paper, Sampson and colleagues investigate how serotonin affects early processing in Drosophila’s visual system. They show that several early processing neurons express different serotonin receptors, which are localized to medulla neurites. They then use bath application of serotonin and serotonin receptor mutations to investigate the functional effects of serotonin on these early visual neurons. The experiments were careful and interpreted in a measured way. Still, there were a few aspects of the research that should be addressed.

Major comments:

These concerns relate to the functional characterization of L2 calcium responses in the presence of serotonin.

1) The presence of TTX in serotonin washes, as the authors also note, is unlikely to isolate cells from the network, since many or most cells in the early visual system don’t spike. Instead, a cocktail of fast neurotransmitter receptor antagonists (against GABA receptors, gluCl, and nicotinic and metabotropic Ach receptors) seems like it would largely do the trick, and eliminate some of the hedging that the current experiments require about cell autonomy of the increased-calcium effect, which appears several times in the manuscript (for instance, Line 486).

2) F6A and F4B show rather different responses to the serotonin and saline, although the genotypes are the same: in F6A the +serotonin trace does not go as high and the saline trace increases, compared to the F4B traces. The increase in the saline condition here but not earlier is especially odd and cannot be explained by the difference in TTX. What explains this variability and how does it affect interpretation of the results?

3) Fig 5D: biggest effect appears to be in rebound amplitude (i.e., the second lobe). Is that statistically significant? Also, it would be important to note in the paper that the indicator itself is a nonlinear function of Ca, so changes you do see can be difficult to attribute to changes in cell activity, rather than changes in Ca-indicator regime. For instance, raising baseline Ca likely changes the gain of this indicator.

4) In F6D, the panels show only differences from the first minute. However, by far the largest differences here appear to exist between the genotypes in epoch 1. These large baseline differences should be explored, and it seems that baseline differences among genotypes make it far more difficult to interpret the changes that occur in the presence of serotonin. This interpretation should also be addressed.

5) Last, I’m not sure I understand the logic of why the +/- would create a “sensitized genetic background”, or even how the experiments support this claim. The comparison made in the paper is between +/- and -/- with serotonin added (Fig 6), compared to +/+ with and without serotonin (Fig 5). I would expect a sensitizing background to be +/- with and without serotonin, which then shows phenotypes when +/+ did not. The comparison of +/- (+ser) with -/- (+ser) seems to have a quite different set of interpretations for the origins of the differences. The paper should make clearer how different these interpretations are in Figures 5 and 6.

Minor comments:

1) Not too much is known about L2 functional properties, but one salient feature is surround inhibition, which was reported to rely on presynaptic inhibition of PRs (Freifeld et al., 2013). Are there experiments to do or speculation to raise about how serotonin might or might not affect this aspect of L2 function, given its actions in other systems? For instance, does raising the baseline calcium make inhibitory conductances drive stronger currents?

2) L5 image in Fig. 1E does not show the medulla axon stratification, which would more convincingly identify L5, if you have such a case. This particular image has a lot of other expression right where one wants to see it. There is not a really good view of axon terminals in the supp fig, either.

3) Enrichment figure S7 – it seems that one might label y=1 as an important point, since that’s potentially a null hypothesis. 0 is not necessarily the null hypothesis with these experiments.

4) Line 578: The first, but not. How about, "This study is the first using the modern genetic toolkit."?

5) As written, the discussion contains a lot of summary of results, rather than building on them or putting them into context. Especially paragraphs 2 and 3. Around line 660, for instance, the text mostly restates points already made in the results.

Reviewer #2: Summary

In this paper, the authors investigate serotonergic processing in the visual periphery of Drosophila. So far, octopamine has been shown to have a functional role in modulating vision, and we know that a range of other amines and peptides are involved, but their roles are unclear. Serotonin has been linked to the visual modulation of circadian circuitry, and the ability to verify such a link and identify biophysical mechanisms involved is an exciting possibility raised by this work.

The authors used MiMIC lines and adult multicolor flip outs to identify neurons expressing serotonin receptors in the optic lamina of Drosophila. They identified that L2 expresses 5-HT2B receptors, T1 expresses 5-HT1A,B, and L5 expresses 5-HT7 (Fig. 1). They also saw expression in other neurons corresponding to C3, Mi1, TmY3 and glia adjacent to the lamina basement membrane, proposed to play a role in modulating the extracellular space of individual lamina cartridges. They confirmed expression of these receptors in L2, and T1 using RNA-seq (Fig. 2). They used targeted GFP expression to confirm that T1 and likely L2 receptor expression is in medulla layer 2, not the lamina (Fig. 3). There was no sybGRASP signal in medulla layer 2, indicating that serotonergic modulation may be paracrine signaling. Bath application of 5-HT increased baseline calcium fluorescence over minutes in the presence of TTX to block excitation via sodium channels in all neurons (Fig 4). Bath application without TTX did not appear to substantially affect responses of L2 neurons to dark (preferred stimulus) flashes, except possibly in the duration of responses (Fig. 5C’). In 5-HT2B-/- mutants, bath application of serotonin had no effect different from saline, but 5-HT2B+/- control flies showed an larger and faster responses to ON and OFF flashes (Fig. 6).

Overall, I am excited to be reading about how serotonin may modulate peripheral visual processing in Drosophila. The paper covers a lot of ground and techniques, and the authors should be proud of having done so much. I have limited my input to my own areas of knowledge. I have concerns about how the imaging data is analyzed and can be interpreted (comments).

The authors tried to link serotonin signaling to rapid visual processing (100 ms stimuli) but they used extremely high bath concentrations to see a modest effect on the signal processing. Rather, the pre-existing link to others aspects of neuromodulation, e.g. circadian regulation to pick one example, would seem a more likely avenue, and one in which modest serotonin levels could alter L2 processing in less pronounced but no less important ways, fitting with their biggest effect seeming to be on baseline calcium levels over minutes.

In its writing, the manuscript lacked clarity in its aims and achievements. The introduction was vague in important places and the discussion very long for what it said. I have given a list of feedback about how the manuscript could press its advantages better in a list of Writing Comments that the authors should feel free to use entirely as they like. The manuscript reads like it has had many review comments already, and the purpose of this section of feedback is to help clarify the message not add to reviewer fatigue, I hope they are helpful.

Comments

• I had issues with how the imaging data is analyzed and presented.

o Why is it not the baseline F in Fig 5A? Clearly the baseline F is varying. I don’t understand why the baseline F can’t be calculated as the F in the few seconds before the stimulus, rather than the mean of the between stimulus activity in epoch 1 (L405). This issue is manifest in Fig 6A where the baseline has a substantial drift with saline. The reason that DF/F is used is because baseline F drifts. This version of DF/F, normalizing to initial F, is odd and is a halfway house between a real-time updated DF/F and baseline F. Fig 6B is referred to as ‘baseline’ (L483) but y-axis is DF/F.

o The duration of the response in 5C’ is more normally measured as the time for the response to decrease from 90%-10%, as the initial peak and final minimum are noisy measurements where the gradient is zero.

o Likewise decay kinetics might ordinarily be measured as exponential fits to 90%-10% decay, with the exponential time scale reported. A linear fit from the peak is odd.

o In 5C it is the difference in peak DF/F before and after serotonin, so the y-axis is not DF/F but (peak DF1/F1 – peakDF2/F2).

• An issue for the imaging part of the paper is the linearity of GCamp6f and GCamp6f dynamics in the presence of substantial changes in intracellular calcium. If the dynamics and range are affected, then the changes in dynamics and amplitude are the result of the change in calcium levels, not the amplitude and dynamics of the short-term responses. I am really not convinced that the amplitude and dynamics are meaningfully affected by 5HT, certainly not compared to apparent shifts in the baseline fluorescence. I have to say, I don’t know how I would address this issue in my own data. In an ideal world, I might check the results using a different reporter, e.g. one of the GCamp6f variants with a displaced linear calcium range – Badura… and Sam Wang 2014 Fig 3 is helpful. To be clear, I am not suggesting you do these experiments, more that the explanation is possible.

• In vivo measurements of serotonin give 280-640 nM, as measured using cyclic voltammetry after Chrimson stimulation of serotoninergic neurons (Borue,… and Jill Venton 2009). I don’t think those measurements are meant to be too stringent a ground truth but 100 uM is greater by x500 or so. This is lower that the 1 mM used by Roger Hardie. Even so, I think there are limits to what we can say about the in vivo effects of 5-HT on signal processing. The paper cites work from Elzbieta Pyza on serotonin as a circadian modulator of L1 and L2, and this seems plausible, especially for T1 which expresses circadian genes eg tim2 (Benna et al 2020 Curr Biol). I think using artificially high hemolymph levels is helpful for interpolating the effects of lower concentrations on slower timescales. I don’t think the paper can argue that serotonin helps L2 encode variable stimuli (implied by Abstract, L16 and use of 100 ms stimuli), but it is helpful and important to identify that serotonin is altering base levels in specific neurons.

• I didn’t find that Fig S3 D gave me the information I needed to identify C2 (mainly I could only see the soma), nor did panel B let me see the L5 medulla expression, which is sort of in Fig 1E but a bit awkwardly on the side. I didn’t read the criteria say in methods by which you identified the cells. I’m sure you are right but occasionally mistakes slip through, like the T2 paper.

• In figure S1, the expression in layer 1 of lobula and outermost layer of lobula plate is interesting. Is this meaningful? Is vertical downwards motion processing affected?

• There is some confusion about TTX, as it is presented as blocking inputs from other cells (L320), which is odd. Being a Na channel blocker, it is blocking depolarization from Na channels, in L2 and other cells. The paper refers to cells ‘unaffected by TTX (L330)’. My understanding is that Drosophila only has para sodium channels, and creates variety by alternative splicing etc, and all variants are blocked by TTX, while in mice there is e.g. Na_v1.9 which is TTX-resistant. The paper discusses that ‘graded potentials’ would be more likely to mediate indirect effects of serotonin in the presence of TTX (L360) as though these might require NA channels. It’s all a bit inconsistent across the paper and odd. The expression of serotonin receptors in glia adjacent to the basement membrane means that there could be a serotinergic regulation of the extracellular ion concentrations in every cartridge, as postulated by Shaw, and investigated and explained in detail by Weckstrom and Laughlin 2010.

Typos

• L30. Without a functional -> without functional

• L30. ‘severe’ reads badly – melodramatic? – but also the differences are in response to the application of unphysiological concentrations of 5-HT.

• L34. ‘…required for initiating visual processing’? Is that not photoreceptors? Clearly photoreceptors process visual signals, e.g. through adaptation.

• L36. …sensory networks -> a sensory network

• L57. Visual processing begins in the retina not the lamina. There is, for instance, the pupil reflex controlling photon capture in the retina, which is even before the photoreceptor.

• L58. They are first order interneurons, the photoreceptors are the first order neurons.

• L61. Identical -> indistinguishable.

• L69. Also in lobula plate.

• L72. Odd that you don’t cite a Nassel paper with Drosophila expression, or mention LBO5HT, which is quite well known. Agata Kołodziejczyk’s 2011 thesis done in his lab did 5HT immunostaining in Drosophila (I think), but I don’t remember seeing it published.

• L95. ‘…crucial for the initiation of visual information processing.’ Initiation? No. Perhaps crucial for early visual processing, but it’s still also so vague – L1 and L2 have well-characterized high bandpass spatiotemporal filters of luminance that reduce temporal and spatial redundancy and enhance contrast detection.

• L126. ‘subtype of cells with a soma’. Hmm. Perhaps ‘subtype of cell’ -> cell type, and ‘with soma’.

• L314. L2split -> L2 split

• L322. ‘Cell-non autonomous’. The paper uses ‘cell non-autonomous’, ‘cell-non autonomous’, and even ‘cell-non-autonomous’.

• L587. The lamina contain a small number of columnar (cartridge neurons) but there is quite a range of cells from accessory medulla etc innervating the lamina. In this paragraph you are repeating material from the introduction and early results and you’d be better discussing your results.

• L1105. Capitalization of journal titles.

• L1118. Osford

Writing comments – these are aimed at lifting the text.

• L15. ‘…highly variable stimulus space.’ You don’t explore how variable stimuli are encoded by L2, so this is an unkind opening sentence for your paper. They have other functions, including circadian modulation, maintaining circuit homeostasis and neuroprotection. These other roles are perhaps more likely landing ground for your paper because they require slow changes in neural activity, such as long lasting changes in baseline calcium.

• L19. ‘Mechanisms by which serotonin regulates visual neurons remain unclear.’ We knew before the paper that visual neurons expressed serotonin receptors and so they likely affected the neurons. The paper shows changes in baseline calcium with very high levels of applied serotonin, and at best minor changes in dynamics. I don’t think that making the mechanisms clearer is the question addressed by this paper. Rather, the paper explicitly addresses hypotheses addressed by other, wide-ranging data-driven studies, and provides evidence that an amine other than octopamine can modulate the physiology of peripheral visual interneurons.

• L19. You use present tense to say what you have done. It makes it unclear which statements are statements of fact, and which are results you have found. The present tense makes it engaging, but you lose clarity in stating achievements.

• L23. ‘…initial steps of visual processing’. Clearly the first visual interneurons are required for early visual processing. This is true for T1 and we have little idea of what it does. Rather, L1 and L2 are involved in detecting spatiotemporal luminance contrast. The photon count hitting photoreceptors covers ten orders of magnitude, but thanks to photoreceptors adaptation, pigment screening in the eye, and processing in the lamina, this is reduced to a voltage range of a few tens of millivolts, which is a stunning compression job.

• L28. Be explicit – decreases the duration of responses is your result, I understood. ‘Alters kinetics’ is very vague and not helpful.

• L33. The phrase non cell-autonomous is used 15 times in the paper and it’s very odd when talking about the cell’s physiological performance embedded in a network of cells. The lamina cells are all highly connected and operating as cartridge ensembles, so the idea that they might be operating cell-autonomously or non-autonomously feels like misplaced emphasis, perhaps inherited from previous work in cell biology. In this sentences you could say ‘…via other cells’ and there wouldn’t be any change in meaning (to this reader).

• L33. The second after the Hardie paper you cite?

• L33. What does ‘functional’ mean? You use it a lot. I think you mean physiological. You don’t in any way identify a function for serotonin, such as improve signal contrast, reduce redundancy etc.

• L35. The ending sentence is very vague. I think if you had written less vague opening sentences you would be able to land the abstract better at the end. My suggestion is to open with roles for serotonin in circadian modulation and maintaining circuit homeostasis, and end with saying that now the physiological basis of these roles can be studied in detail.

• L39. ‘…not known.’ But linked to circadian function, as you cite. If you are going to identify a knowledge gap, you want to set out why it would be interesting to fill the gap.

• L42. ‘…can alter…’ too vague again, be explicit.

• L42. See earlier comments on being ‘first’ and ‘functional’.

• L47-50. Your examples focus on signal processing, but the main focus of the retina papers you cite is on eye development and neuroprotection against excitotoxicity. Why focus on signal processing when you don’t analyze anything beyond ON and OFF flashes, and the biggest effect is in changes in baseline calcium?

• L52. ‘…can modulate the response…’ be explicit about the changes you report.

• L55. What does powerful mean? It’s an intensifier without qualification, be explicit. If it means there are lots of tools, then say that.

• L85. Vague! Of course receptors can do different things. Give examples, make it grounded.

• L87. ‘…it is likely that…’ Of course activity in one cell can propagate to another. I think you are trying to set the reader up for the fact that L1 is affected by adding 5HT later, but this does not help.

• L91. ‘…affects visual processing…’ Vague!

• L93. ‘functional effect’? See earlier comments.

• L113. ‘prominent role’? If they don’t have the lamina they are blind. Is this what ‘prominent role’ means? The lamina compresses ten orders of magnitude of photon counts into 10s of mV, and uses the superposition principle to enhance luminance sensitivity, as well as a myriad of neural and non-neural mechanisms. Be explicit.

• L116. ‘…previously characterized.’ But T1 hasn’t as you have pointed out. Rather, if serotonin is important for circadian input, it means you should look in the lamina, a brain region well innervated by serotinergic processes from e.g. the accessory medulla.

• L120. ‘Importantly,…’ Repetition from L112. In any case, the paper is very tentative in its classification of cell types, rather than confidently stating e.g. L5.

• L123. ‘Although MCFO…’ belongs in methods. It reads like a riposte to a reviewer.

• L125. ‘Using…’ Formulation of this sentence is repetitive of L119-120.

• L141. I really didn’t like that every result is immediately compared to Ref2 61 and 62. It makes for a very slow reading experience. Can you not flag refs 61 and 62 at the start and say that you will discuss the comparison at the end of the section? The paper deserves to get a clear run at stating its own results.

• L161. ‘As noted above’. I looked and couldn’t find it. Recommend you delete this sentence.

• L191-201. This is all discussion and not results. Maybe a couple of sentences is fine, but this long stretch diluted from the impact of the results.

• L203. ‘Before embarking…’ Recommend delete this opening phrase. ‘We sought to confirm…’

• 213. The conventional shortening is ‘std’, no? stdev is odd.

• L312. It’s kind to say what the G_q/11 is, a G protein alpha subunit.

• L319. You should say that TTX is a Na channel blocker here, not later.

• L397. It’s a little bit odd to use ref 31 as the reference for L1 and L2 calcium dynamics, as though that is the first or best paper to describe the properties of L1 and L2.

• L417. ‘rebound’ is a bit of a loaded way to describe it. It is an OFF response to the end of the ON stimulus. Rebound is usually used for mechanisms that are activated by hyperpolarization, and it is not clear that this is the case in the L2 calcium response.

• L432. Figure legend is far too long. The point of the legend is just to enable the reader to read the figures, not explain them, give methods etc – that’s the job of the main paper.

• L477. Can you ‘blunt’ a ‘gradual increase’? Maybe ‘diminish’?

• L523. Figure legend is far too long again, very repetitive with Figure 5. You can say that the same conventions are used, where appropriate.

• L569. 7 pages is a huge Discussion for this paper. Maybe 5 max?

• L572. Functional -> physiological, and same comment throughout Discussion.

• L578. ‘…to determine their potential function.’ You didn’t determine the function, so it makes the paper look weak to say you did. I think I have tried to give suggestions about what you have done above.

• L579. Saying you are the first after the first is very tortured – you are the second study. Nothing shameful about that and you can be proud.

• L616. Excellent! You handle this very well. You come out saying you did not find L3 expression, and since L3 expression was so strong in Davis et al, it’s very useful. Their L3 split has some expression of Lawf2, and that may explain the shared expression of 5-HT1A/B in Lawf2 and L3.

• L686. The ultrastructure is knowable now, in that it is available to trace in the FAFB data set.

• L700-705. Repetition! Why am I reading this introduction to L1 and L2 here?

• L748. The last thing I read is in present tense, saying what you have done – very odd.

• L981. Calcium imaging was performed as previously described, but reference is to the Keles and Frye paper and Keles is not an author, so it’s odd.

• The methods were very nicely written – a nice clear, explicit and direct way of writing. I would have appreciated seeing examples of ROIs.

Reviewer #3: In the current study, the authors conduct a good standard research to prodvide how serotonergic signaling processes sensory information. Using a combination of MiMICs, MCFO, FACS, RNA-seq and RT-qPCR, the authors maped the expression patterns of 5-HT receptor neurons in the visual system, and found that 5-HT2B receptors are expressed in lanima monopolar cell L2; 5-HT1A and 5-HT1B are epressed in T1 cells; and 5-HT7 is expressed in L5 monopolar cells of the lanima. In addition, the authors observed 5-HT2A cells exhibit morpholgic similarity to optic glia cells. Furthermore, 5-HT2B and 5-HT1A receptors localize to layer M2 of the medulla. Functionally, the authors provide evidence of the initial visual processing cell subtypes as well as the primary site of 5-HT regualtion by in vivo calcium imaging.

There are several points that need to be addressed:

1) There are multiple receptors types locating to subtypes of cells in the visual system, however, where the 5-HT signal comes from is not known from the current study.

2) A previous study (Konstantinides 2018) employed FACS-SMART-seq to probe different receptors in the visual system, so as the current study. However, the resutls have a few overlaps. The authors discussed that the differences might be due to the mehods used to isolate target cell populations, but it would be better to have at least some clarification of the expression levels in the C2 neurons of the lamina neurons to reconcile the differences, since there are 3 types of 5-HT receptors reported in the previous study and none were detected.

3) Typos: Line 383 "decreased in the light in"

Line 607 "in the current study, Davis et al, and Konstantinides et al." unfinished sentence.

**Have all data underlying the figures and results presented in the manuscript been provided?**

Reviewer #1: No: I could not find a data availability statement in the methods describing where data was deposited, for instance calcium imaging data or sequence data. The authors filled out that the data was all available, so this may have been an oversight or I could have missed it.

Reviewer #2: Yes

Reviewer #3: Yes

PLOS authors have the option to publish the peer review history of their article (what does this mean?). If published, this will include your full peer review and any attached files.

Reviewer #1: No

Reviewer #2: No

Reviewer #3: No

---

## [Editor Report · Decision Letter 1]

22 Jul 2020

Dear David,

We are pleased to inform you that your manuscript entitled "Serotonergic modulation of visual neurons in Drosophila melanogaster" has been editorially accepted for publication in PLOS Genetics. Congratulations!

Yours sincerely,

Claude

Claude Desplan

Associate Editor

PLOS Genetics

Gregory P. Copenhaver

Editor-in-Chief

PLOS Genetics

Comments from the reviewers (if applicable):

**Data Deposition**

http://datadryad.org/submit?journalID=pgenetics&manu=PGENETICS-D-20-00681R1

**Press Queries**

---

## [Editor Report · Acceptance letter]

24 Aug 2020

PGENETICS-D-20-00681R1 

Serotonergic modulation of visual neurons in Drosophila melanogaster 

Dear Dr Krantz, 

We are pleased to inform you that your manuscript entitled "Serotonergic modulation of visual neurons in Drosophila melanogaster" has been formally accepted for publication in PLOS Genetics! Your manuscript is now with our production department and you will be notified of the publication date in due course.

With kind regards,

Kaitlin Butler

PLOS Genetics

On behalf of:
